# Foundation Inference Models for Markov Jump Processes

**David Berghaus[1,2], Kostadin Cvejoski[1,2], Patrick Seifner[1,3]**
**César Ojeda[4] & Ramsés J. Sánchez[1,2,3]**
Lamarr Institute[1], Fraunhofer IAIS[2], University of Bonn[3] & University of Potsdam[4]
`{david.berghaus, kostadin.cvejoski}@iais.fraunhofer.de`
`seifner@cs.uni-bonn.de, ojedamarin@uni-potsdam.de, sanchez@cs.uni-bonn.de`

## Abstract

Markov jump processes are continuous-time stochastic processes which describe dynamical systems evolving in discrete state spaces. These processes find wide application in the natural sciences and machine learning, but their inference is known to be far from trivial. In this work we introduce a methodology for *zero-shot inference* of Markov jump processes (MJPs), on bounded state spaces, from noisy and sparse observations, which consists of two components. First, a broad probability distribution over families of MJPs, as well as over possible observation times and noise mechanisms, with which we simulate a synthetic dataset of hidden MJPs and their noisy observations. Second, a neural recognition model that processes subsets of the simulated observations, and that is trained to output the initial condition and rate matrix of the target MJP in a supervised way. We empirically demonstrate that *one and the same* (pretrained) recognition model can infer, *in a zero-shot fashion*, hidden MJPs evolving in state spaces of different dimensionalities. Specifically, we infer MJPs which describe (i) discrete flashing ratchet systems, which are a type of Brownian motors, and the conformational dynamics in (ii) molecular simulations, (iii) experimental ion channel data and (iv) simple protein folding models. What is more, we show that our model performs on par with state-of-the-art models which are trained on the target datasets.

Our pretrained model, repository and tutorials are available online[1].

## 1 Introduction

Very often one encounters dynamic phenomena of wildly different nature, that display features which can be reasonably described in terms of a macroscopic variable that jumps among a finite set of *long-lived*, metastable discrete states. Think, for example, of the changes in economic activity of a country, which exhibit jumps between recession and expansion states (Hamilton, 1989), or the internal motion in proteins or enzymes, which feature jumps between different conformational states (Elber and Karplus, 1987). The states in these phenomena are said to be long-lived, inasmuch as every jump event among them is rare, at least as compared to every other event (or subprocess, or fluctuation) that composes the phenomenon and that occurs, by construction, *within* the metastable states. Such a description in terms of macroscopic variables effectively decouples the fast, intra-state events from the slow, inter-state ones, and allows for a simple probabilistic treatment of the jumping sequences as Markov stochastic processes: the *Markov Jump Processes* (MJPs). In this work we are interested in the general problem of inferring the MJPs that best describe empirical (time series) data, recorded from dynamic phenomena of very different kinds.

---

[1] https://fim4science.github.io/OpenFIM/intro.html

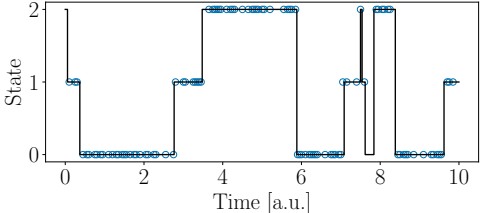 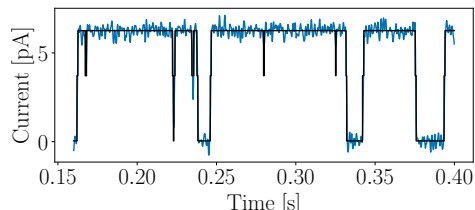

Figure 1: Processes of very different nature (seem to) feature similar jump processes. *Left*: State values (blue circles) recorded from the discrete flashing ratchet process (black line). *Right*: Current signal (blue line) recorded from the viral potassium channel $Kcv_{MT35}$, together with one possible coarse-grained representation (black line).

To set the stage, let us assume that we want to study some $D$-dimensional empirical process $\mathbf{z}(t) : \mathbb{R}^+ \rightarrow \mathbb{R}^D$, which features long-lived dynamic modes, trapped in some discrete set of metastable states. Let us call this set $\mathcal{X}$. Let us also assume that we can obtain a macroscopic, coarse-grained representation from $\mathbf{z}(t)$ — say, with a clustering algorithm — in which the fast, intra-state events have been integrated out (*i.e.* marginalized). Let us call this macroscopic variable $X(t) : \mathbb{R}^+ \rightarrow \mathcal{X}$. If we now make the Markov assumption and define the quantity $f(x|x')\Delta t$ as the infinitesimal probability of observing one jump from state $x'$ (at some time $t$), into a different state $x$ (at time $t + \Delta t$), we can immediately write down, following standard arguments (Gardiner, 2009), a differential equation that describes the probability distribution $p_{MJP}(x, t)$, over the discrete set of metastable states $\mathcal{X}$, which encapsulates the state of the process $X(t)$ as time evolves, that is

$$\frac{dp_{MJP}(x,t)}{dt} = \sum_{x' \neq x} \Big( f(x|x')p_{MJP}(x',t) - f(x'|x)p_{MJP}(x,t) \Big). \tag{1}$$

Equation 1 is the so-called *master equation* of the MJP whose solutions are completely characterized by an initial condition $p_{MJP}(x, t = 0)$ and the transition rates $f : \mathcal{X} \times \mathcal{X} \rightarrow \mathbb{R}^+$.

With these preliminaries in mind, we shall say that to infer an MJP from a set of (noisy) observations $\mathbf{z}(\tau_1), \dots, \mathbf{z}(\tau_l)$ on the process $\mathbf{z}(t)$, recorded at some observation times $\tau_1, \dots, \tau_l$, means to infer both the transition rates and the initial condition determining the *hidden* MJP $X(t)$ that best explains the observations. In practice, statisticians typically assume that they directly observe the coarse-grained process $X(t)$. That is, they assume they have access to the (possibly noisy) values $x_1, \dots, x_l$, taken by $X(t)$ at the observation times $\tau_1, \dots, \tau_l$ (see Section 2). We shall start from the same assumptions. Statisticians then tackle the inference problem by (i) defining some (typically complex) model that encodes, in one way or the other, equation 1 above; (ii) parameterizing the model with some trainable parameter set $\theta$; and (iii) updating $\theta$ to fit the empirical dataset.

One issue with this approach is that it turns the inference of hidden MJPs into an instance of an *unsupervised learning problem*, which, as history shows, is far from trivial (see Section 2). Another major issue is that, if one happens to succeed in training said model, the trained parameter set $\theta^*$ will usually be overly specific to the training set $\{(x_1, \tau_1), \dots, (x_l, \tau_l)\}$, which means it will likely struggle to handle a second empirical process, even if the latter can be described by a similar MJP. Figure 1 contains snapshots from two empirical processes of very different nature. The figure on the left shows a set of observations (blue circles) recorded from the discrete flashing ratchet process (black line). The figure on the right shows the ion flow across a cell membrane, which jumps between different activity levels (blue line). Despite the vast differences between the physical mechanisms underlying each of these processes, the coarse-grained representations of the second one (black line) is abstract enough to be strikingly similar to the first one. Now, we expect that — at this level of representation — one could train *a single inference model to fit each process* (separately). Unfortunately, we also expect that an inference model trained to fit *only one* of these (coarse-grained) processes, will have a hard time describing the second one.

In this paper we will argue that the *notion of an MJP description* (in coarse-grained space) is simple enough, that it can be encoded into the weights of a single neural network model. Indeed, instead of training, in an unsupervised manner, a complex model (which somehow encodes the

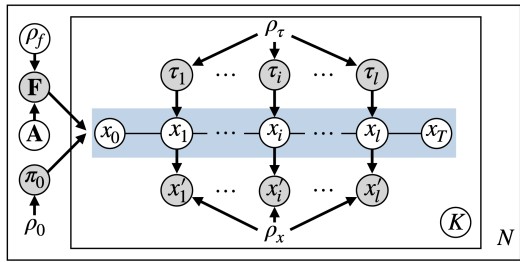 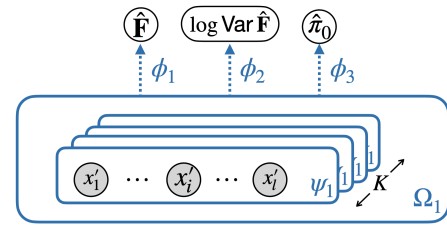

Figure 2: Foundation Inference Model (FIM) for MJP. *Left*: Graphical model of the FIM (synthetic) data generation mechanism. Filled (empty) circles represent observed (unobserved) random variables. The light-blue rectangle represents the continuous-time MJP trajectory, which is observed discretely in time. See main text for details regarding notation. *Right*: Inference model. The network $\psi_1$ is called $K$ times to process $K$ different time series. Their outputs is first processed by the attention network $\Omega_1$ and then by the FNNs $\phi_1$, $\phi_2$ and $\phi_3$ to obtain the estimates $\hat{\mathbf{F}}$, $\log \text{Var} \, \hat{\mathbf{F}}$ and $\hat{\boldsymbol{\pi}}_0$, respectively.

master equation) on a single empirical process; we will train, in a supervised manner, a simple neural network model on a *synthetic dataset that is composed of many different MJPs, and hence implicitly encodes the master equation*. This procedure can be understood as an *amortization* of the probabilistic inference process through a single *recognition model*, and is therefore akin to the works of Stuhlmüller et al. (2013), Heess et al. (2013) and Paige and Wood (2016). Rather than treating, as these previous works do, our (pretrained) recognition model as auxiliary to Monte Carlo or expectation propagation methods, we employ it to directly infer hidden MJPs from various synthetic, simulation and experimental datasets, *without any parameter fine-tuning*. We thus adopt the "zero-shot" terminology introduced by Larochelle et al. (2008), by which we mean that our procedure aims to recognize objects (i.e. MJPs) whose instances (i.e. noisy and sparse series of observations on them) may have not been seen during training. We have recently shown that such an amortization can be used to train a recognition model to perform *zero-shot imputation* of time series data (Seifner et al., 2024). Below we demonstrate that it can also be used to train a model of minimal inductive biases, to perform *zero-shot inference* of hidden MJPs from empirical processes of very different kinds, which take values in state spaces of different sizes. We shall call this recognition model *Foundation Inference Model*[2] (FIM) for Markov jump processes.

In what follows, we first review both classical and recent solutions to the MJP inference problem in Section 2. We then introduce the FIM methodology in Section 3, which consists of a synthetic data generation model and a neural recognition model. In Section 4 we empirically demonstrate that our methodology is able to infer MJPs from a discrete flashing ratchet process, as well as from molecular dynamics simulations and experimental ion channel data, all in a zero-shot fashion, while performing on par with state-of-the-art models which are trained on the target datasets. Finally, Section 5 closes the paper with some concluding remarks about future work, while Section 6 comments on the main limitations of our methodology.

## 2    Related Work

The inference of MJP from noisy and sparse observations (in coarse-grained space) is by now a classical problem in machine learning. There are three main lines of research. The first (and earliest) one attempts to directly optimize the MJP transition rates, to maximize the likelihood of the discretely observed MJP via expectation maximization (Asmussen et al., 1996; Bladt and Sørensen, 2005; Metzner et al., 2007). Thus, these works encode the MJP inductive bias directly into their architecture. The second line of research leverages a Bayesian framework to infer the posterior distribution over the transition rates, through various Markov chain Monte Carlo (MCMC) algorithms (Boys et al., 2008; Fearnhead and Sherlock, 2006; Rao and Teg, 2013; Hajiaghayi et al., 2014). Accordingly, these

---

[2]We name our model foundation model because it aligns with the definition proposed by Bommasani et al. (2021). Indeed, they define foundation models as any model that is trained on broad data (generally using self-supervision at scale) that can be adapted to a wide range of downstream tasks.

simulation-based approaches encode the MJP inductive bias directly into their trainable sampling distributions. The third one, also Bayesian in character, involves variational inference. Within it, one finds again MCMC (Zhang et al., 2017), as well as expectation maximization (Opper and Sanguinetti, 2007) and moment-based (Wildner and Koeppl, 2019) approaches. More recently, Seifner and Sánchez (2023) used neural variational inference (Kingma and Welling, 2013) and neural ODEs (Chen et al., 2018) to infer an implicit distribution over the MJP transition rates. All these variational methods encode the MJP inductive bias into their training objective and, in some cases, into their architecture too.

Besides the model of Seifner and Sánchez (2023), which automatically infers the coarse-grained representation $X(t)$ from $D$-dimensional, countinuous signals, all the solutions above tackle the MJP inference problem directly in coarse-grained space. Yet below, we also investigate the conformational dynamics of physical systems for which the recorded data lies in a continuous space. To approach such type of problems, we will first need to define a coarse-grained representation of the state space of interest. Fortunately for us, there is a large body of works, within the molecular simulation community, precisely dealing with different methods to obtain such representations, and we refer the reader to *e.g.* Noé et al. (2020) for a review. McGibbon and Pande (2015), for example, leveraged one such method to infer the MJP transition rates describing a molecular dynamics simulation via maximum likelihood. Alternatively, researchers have also treated the conformational states in these systems as core sets, and inferred phenomenological MJP rates from them (Schütte et al., 2011), or modelled the fast intra-state events as diffusion processes, indexed by a hidden MJP, and inferred the latter either via MCMC (Kilic et al., 2021; Köhs et al., 2022) or variational (Horenko et al., 2006; Köhs et al., 2021) methods.

In this work we tackle the classical MJP inference problem *on coarse-grained space* and present, to the best of our knowledge, its first zero-shot solution.

## 3 Foundation Inference Models

In this section we introduce a novel methodology for zero-shot inference of Markov jump processes which frames the inference task as a supervised learning problem. Our main assumption is that the space of *realizable MJPs*[3], which take values on bounded state spaces that are not too large, is simple enough to be covered by a heuristically constructed synthetic distribution over noisy and discretely observed MJPs. If this assumption were to hold, a model trained to infer the hidden MJPs within a synthetic dataset sampled from this distribution *would automatically perform zero-shot inference on any unseen sequence of empirical observations*. We do not intend to formally prove this assumption. Rather, we will empirically demonstrate that a model trained in such a way can indeed perform zero-shot inference of MJPs in a variety of cases.

Our methodology has two components. First, a data generation model that encodes our believes about the class of realizable MJPs we aim to model. Second, a neural recognition model that maps subsets of the simulated MJP observations onto the initial condition and rate matrix of their target MJPs. We will explore the details of these two components in the following sections.

### 3.1 Synthetic Data Generation Model

In this subsection we define a broad distribution over possible MJPs, observation times and noise mechanisms, with which we simulate an ensemble of noisy, discretely observed MJPs. Before we start, let us remark that we will slightly abuse notation and denote both probability distributions and their densities with the same symbols. Similarly, we will also denote both random variables and their values with the same symbols.

Let us denote the size of the largest state space we include in our ensemble with $C$, and arrange all transition rates, for every MJPs within the ensemble, into $C \times C$ rate matrices. Let us label these matrices with $\mathbf{F}$. We define the probability of recording the noisy sequence $x'_1, \ldots, x'_l \in \mathcal{X}$, at the

---

[3]By realizable MJPs we mean here MJPs that can be inferred from physical processes, given the typical experimental constraints, like *e.g.* temporal or spatial resolution.

observation times $0 < \tau_1 < \cdots < \tau_l < T$, with $T$ the observation time horizon, as follows

$$\prod_{i=1}^{l} p_{\text{noise}}(x_i'|x_i, \rho_x) p_{\text{MJP}}(x_i|\tau_i, \mathbf{F}, \pi_0) p_{\text{grid}}(\tau_1, \ldots, \tau_l|\rho_\tau) p_{\text{rates}}(\mathbf{F}|\mathbf{A}, \rho_f) p(\mathbf{A}, \rho_f) p(\pi_0|\rho_0). \quad (2)$$

Next, we specify the different components of Eq. 2, starting from the right.

**Distribution over initial conditions**. The distribution $p(\pi_0|\rho_0)$, with hyperparameter $\rho_0$, is defined over the $C$-simplex, and encodes our beliefs about the initial state (*i.e.* the preparation) of the system. It enters the master equation as the class probabilities of the *categorical distribution* over the states of the system, at the start of the process. That is $p_{\text{MJP}}(x, t = 0) = \text{Cat}(\pi_0)$. We either choose $\pi_0$ to be the class probabilities of the stationary distribution of the process, or sample it from a Dirichlet distribution. Appendix B provides the specifics.

**Distribution over rate matrices**. The distribution $p_{\text{rates}}(\mathbf{F}|\mathbf{A}, \rho_f)$ over the rate matrices encodes our beliefs about the class of MJPs we expect to find in practice. We define it to cover MJPs with state spaces whose sizes range from 2 until $C$, because we want our FIM to be able to handle processes taking values in all those spaces. The distribution is conditioned on the adjacency matrix $\mathbf{A}$, which encodes only connected state spaces (*i.e.* irreducible embedded Markov chains only), and a hyperparameter $\rho_f$ which encodes the range of rate values within the ensemble. Specifically, we define the transition rates as $F_{ij} = a_{ij} f_{ij}$, where $a_{ij}$ is the corresponding entry of $\mathbf{A}$ and $f_{ij}$ is sampled from a set of Beta distributions, with different hyperparameters $\rho_f$. Note that these choices restrict the values of the transition rates within the ensemble to the interval $(0, 1)$ and hence, they restrict the number of *resolvable transitions* within the time horizon $T$ of the simulation. We refer the reader to Appendix B, where we specify the prior $p(\mathbf{A}, \rho_f) = p(\mathbf{A})p(\rho_f)$ and its consequences, as well as give details about the sampling procedure. We also discuss the main limitations of choosing a Beta prior over the transition rates in Section 6.

**Distribution over observation grids**. The distribution $p_{\text{grid}}(\tau_1, \ldots, \tau_l|\rho_\tau)$, with hyperparameter $\rho_\tau$, gives the probability of observing the MJP at the times $\tau_1, \ldots, \tau_l$, and thus encodes our uncertainty about the recording process. Given that we do not know a priori whether the data will be recorded regularly or irregularly in time, nor we know its recording frequency, we define this distribution to cover both regular and irregular cases, as well as various recording frequencies. Note that the number of observation points on the grid is variable. Please see Appendix B for details.

**Distribution over noise process**. Just as the (instantaneous) solution of the master equation $p_{\text{MJP}}(x|t, \mathbf{F}, \pi_0)$, the noise distribution $p_{\text{noise}}(x'|x, \rho_x)$, with hyperparameter $\rho_x$, is defined over the set of metastable states $\mathcal{X}$. Recall that FIM solves the MJP inference problem directly in coarse-grained space. The noise distributions then encodes both, possible measurement errors that propagate through the coarse-grained representation, or noise in the coarse-grained representation itself. We provide details of its implementation in Appendix B.

We use the generative model, Eq. 2 above, to generate $N$ MJPs, taking values on state spaces with sizes ranging from 2 to $C$. We then sample $K$ paths per MJP, with probability $p(K)$, on the interval $[0, T]$. The $j$th instance of the dataset thus consists of $K$ paths and is given by

$$\mathbf{F}_j \sim p_{\text{rates}}(\mathbf{F}|\mathbf{A}_j, \rho_{fj}), \text{ and } \pi_{0j} \sim p(\pi_0|\rho_0), \text{ with } (\mathbf{A}_j, \rho_{fj}) \sim p(\mathbf{A}, \rho_f),$$

$$\text{so that } \left\{ X_{jk}(t) \right\}_{k=1}^{K} \sim \text{Gillespie}(\mathbf{F}_j, \pi_{0j}), \quad (3)$$

$$\text{and } \left\{ x_{jki}' \sim p_{\text{noise}}(x'|X_{jk}(\tau_{jki})) \right\}_{(k,i)=(1,1)}^{(K,l)}, \text{ with } \left\{ \tau_{jk1}, \ldots, \tau_{jkl} \right\}_{k=1}^{K} \sim p_{\text{grid}}(\tau_1, \ldots, \tau_l|\rho_\tau),$$

where Gillespie denotes the Gillespie algorithm we use to sample the MJP paths (see Algorithm 1). Note that we make the number of paths ($K$ above) per MJP random, because we do not know a priori how many realizations (*i.e.* experiments), from the empirical process of interest, will be available at the inference time. We refer the reader to Appendix B for additional details.

Figure 2 illustrates the complete data generation process.

## 3.2 Supervised Recognition Model

In this subsection we introduce a neural recognition model that processes a set of $K$ time series of the form $\{(x_{k1}', \tau_{k1}), \ldots, (x_{kl}', \tau_{kl})\}_{k=1}^{K}$, as generated by the procedure in Eq. 3 above, and estimates

the intensity rate matrix $\mathbf{F}$ and initial distribution $\boldsymbol{\pi}_0$ of the hidden MJP. Practically speaking, we would like the model to be able to infer MJPs from time series with observation times *on any scale*. To ensure this, we first normalize all observation times to lie on the unit interval, by dividing them by the maximum observation time $\tau_{\max} = \max\{\tau_{k1}, \ldots, \tau_{kl}\}_{k=1}^{K}$, and then rescale the output of the model accordingly (see Appendix C for details).

Let us use $\phi$, $\psi$ and $\Omega$ to denote feed-forward, sequence processing networks, and attention networks, respectively. Thus $\psi$ can denote *e.g.* LSTM or Transformer networks, while $\Omega$ can denote *e.g.* a self-attention mechanism. Let us also denote the networks' parameters with $\theta$.

We first process each time series with a network $\psi_1$ to get a set of $K$ embeddings, which we then summarize into a global representation $\mathbf{h}_\theta$ through the attention network $\Omega_1$. In equations, we write

$$\mathbf{h}_\theta = \Omega_1(\mathbf{h}_{1\theta}, \ldots, \mathbf{h}_{K\theta}, \theta) \ \text{ with } \ \mathbf{h}_{k\theta} = \psi_1(x'_{k1}, \tau_{k1}, \ldots, x'_{kl}, \tau_{kl}, \theta) \text{ and } k = 1, \ldots, K. \quad (4)$$

Next we use the global representation to get an estimate of the intensity rate matrix, which we artificially model as a Gaussian variable with positive mean, and the initial distribution of the hidden MJP as follows

$$\hat{\mathbf{F}} = \exp(\phi_1(\mathbf{h}_\theta, \theta)), \quad \operatorname{Var}\hat{\mathbf{F}} = \exp(\phi_2(\mathbf{h}_\theta, \theta)) \ \text{ and } \ \hat{\boldsymbol{\pi}}_0 = \phi_3(\mathbf{h}_\theta, \theta), \quad (5)$$

where the exponential function ensures the positivity of our estimates, and the variance is used to represent the model's *uncertainty* in the estimation of the rates (Seifner et al., 2024). The right panel of Figure 2 summarizes the recognition model, and Appendix C provides additional information about the inputs to, outputs of and rescalings done by the model.

**Training objective**. We train the model to maximize the likelihood of its predictions, taking care of the exact zeros (*i.e.* the missing links) in the data. To wit

$$
\begin{aligned}
\mathcal{L} \ = \ &- \mathop{\mathbb{E}}_{\mathbf{F}, \mathbf{A} \sim p_{\text{rates}}} \left\{ \sum_{ij=1}^{C} a_{ij} \Big[ \frac{(f_{ij} - \hat{f}_{ij})^2}{2\operatorname{Var}\hat{f}_{ij}} + \frac{1}{2} \log \operatorname{Var}\hat{f}_{ij} \Big] - \lambda(1 - a_{ij}) \Big[ \hat{f}_{ij}^2 + \operatorname{Var}\hat{f}_{ij} \Big] \right\} \\
&- \mathop{\mathbb{E}}_{\boldsymbol{\pi}_0 \sim p} \left\{ \sum_{i=1}^{C} \pi_{i0} \log \hat{\pi}_{i0} \right\},
\end{aligned}
\quad (6)
$$

where the second term is nothing but the mean-squared error of the predicted rates $\hat{f}_{ij}$ (and its standard deviation) when the corresponding link is missing, and can be understood as a regularizer with weight $\lambda$. The latter is a hyperparameter.

**FIM context number**. During training, FIM processes a variable number $K$ of time series, which lies on the interval $[K_{\min}, K_{\max}]$. Similarly, each one of these time series has a variable number $l$ of observation points, which lies on the interval $[l_{\min}, l_{\max}]$. We shall say that FIM needs a bare minimum of $K_{\min}l_{\min}$ input data points to function. Perhaps unsurprisingly, we have empirically seen that FIM perform bests when processing $K_{\max}l_{\max}$ data points. Going significantly beyond this number seems nevertheless to decrease the performance of FIM. We invite the reader to check Appendix D for details.

Let us define then, for the sake of convenience, the FIM context number $c(K, l) = Kl$ as the number of input points[4] FIM makes use of to estimate $\mathbf{F}$ and $\boldsymbol{\pi}_0$.

## 4 Experiments

In this section we test our methodology on five datasets of varying complexity, and corrupted by noise signals of very different nature, whose hidden MJPs are known to take values in state spaces of different sizes. In what follows we use *one and the same* (pretrained) FIM to infer hidden MJPs from all these datasets, *without any parameter fine-tuning*. Our FIM was (pre)trained on a dataset of 45K MJPs, defined over state spaces whose sizes range from 2 to 6. A maximum of ($K =$)300 realizations (paths) *per MJP* were observed during training, everyone of which spanned a time-horizon $T = 10$, recorded at a maximum of 100 time points, 1% of which were mislabeled. Given these specifications, FIM is expected to perform best for the context number $c(300, 100)$ during evaluation. Additional

---

[4] We can think about it as the context length in large language models.

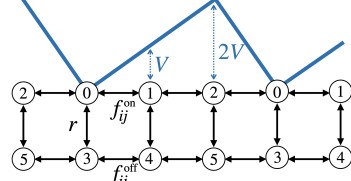

Figure 3: Illustration of the six-state discrete flashing ratchet model. The potential $V$ is switched on and off at rate $r$. The transition rates $f_{ij}^{\text{on}}, f_{ij}^{\text{off}}$ allow the particle to propagate through the ring.

|  | $V$ | $r$ | $b$ |
|---|---|---|---|
| GROUND TRUTH | 1.00 | 1.00 | 1.00 |
| NEURALMJP | **1.06** | 1.17 | 1.14 |
| FIM | 1.11(7) | **0.99(8)** | **0.98(5)** |

Table 1: Inference of the discrete flashing ratchet process. The FIM results correspond to FIM evaluations with context number $c(300, 50)$, averaged over 15 batches.

information regarding model architecture, hyperparameter selection and other training details can be found in Appendix D.

**Baselines**: Depending on the dataset, we compare our findings against the NeuralMJP model of Seifner and Sánchez (2023), the switching diffusion model (SDiff) of Köhs et al. (2021), and the discrete-time Markov model (VampNets) of Mardt et al. (2017).

All these baselines are trained on the target datasets.

### 4.1 The Discrete Flashing Ratchet (DFR): A Proof of Concept

In statistical physics, the ratchet effect refers to the rectification of thermal fluctuations into directed motion to produce work, and goes all the way back to Feynman (Feynman et al., 1965). Here we consider a simple example thereof, in which a Brownian particle, immersed in a thermal bath at unit temperature, moves on a one-dimensional lattice. The particle is subject to a linear, periodic and asymmetric potential of maximum height $2V$ that is switched on and off at a constant rate $r$. The potential has three possible values when is switched on, which correspond to three of the states of the system. The particle jumps among them with rate $f_{ij}^{\text{on}}$. When the potential is switched off, the particle jumps freely with rate $f_{ij}^{\text{off}}$. We can therefore think of the system as a six-state system, as illustrated in Figure 3. Similar to Roldán and Parrondo (2010), we now define the transition rates as

$$f_{ij}^{\text{on}} = \exp\left(-\frac{V}{2}(j - i)\right), \text{ for } i, j \in (0, 1, 2); \quad f_{ij}^{\text{off}} = b, \text{ for } i, j \in (3, 4, 5). \quad (7)$$

Given these specifics, we consider the parameter set $(V, r, B) = (1, 1, 1)$ together with the dataset simulated by Seifner and Sánchez (2023), which consists of 5000 paths (in coarse-grained space) recorded on an irregular grid of 50 time points. The task is to infer $(V, r, B)$ from these time series. NeuralMJP infers a *global* distribution over the rate matrices and hence relies on their entire train set, which amounts to about 4500 time series. We therefore report FIM evaluations with context number $c(300, 50)$ on that same train set, averaged over 15 (non-overlapping) batches in Table 1.

The results show that FIM performs on par with (or even better than) NeuralMJP, *despite not having been trained on the data*. Note in particular that our results are sharply peaked around their mean, indicating that a context of $c(300, 50)$ points only contains enough information to describe the data well. What is more, Table 16 in the Appendix demonstrates that FIM can infer vanishing transition rates as well (see Eq. 6). Now, being able to infer the rate matrix in zero-shot mode allows us to immediately estimate a number of observables of interest *without any training*. Stationary distributions, relaxation times and mean first-passage times (see Appendix A for their definition), as well as time-dependent moments, can all be computed zero-shot via FIM. For example, we report on the left block of Figure 4 the time-dependent class probabilities (*i.e.* the master eq. solutions) computed with the FIM-inferred rate matrix (black), against the ground-truth solution (blue). The agreement is very good.

**Zero-shot estimation of entropy production**. The DFR model is interesting because the random switching combined with the asymmetry in the potential make it more likely for the particle to jump towards the right (see Figure 4). Indeed, that is the ratchet effect. As a consequence, the system features a stationary distribution with a net current — the so-called *non-equilibrium steady state*

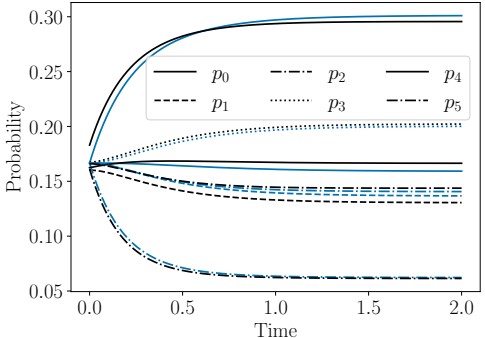 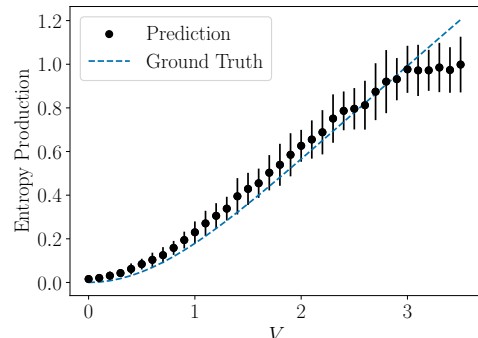

Figure 4: Zero-shot inference of DFR process. *Left*: master eq. solution $p_{\text{MJP}}(x, t)$ as time evolves, wrt. the (averaged) FIM-inferred rate matrix is shown in black. The ground-truth solution is shown in blue. *Right*: Total entropy production computed from FIM (over a time-horizon $T = 2.5\,[a.u.]$). The model works remarkably well for a *continuous range* of potential values.

(Ajdari and Prost, 1992), which is characterized by a non-vanishing (stochastic) entropy production. The development of (neural) estimators of entropy production is a very active topic of current research (see *e.g.* Kim et al. (2020) and Otsubo et al. (2022)). Given that the entropy production can be written down in closed form as a function of both the rate matrix and the master eq. solution (see *e.g.* Seifert (2012)), we can readily use FIM to estimate it.

Figure 4 displays the total entropy production computed with FIM for a set of different potentials. The results are averaged over 15 FIM evaluations with $c(300, 50)$ and are again in very good agreement with the ground truth. It is noteworthy that FIM, trained on our heuristically constructed dataset, captures well *a continuous set of MJPs*. That is, we evaluate *one and the same* FIM over different datasets, each sampled from a DFR model with a different potential value. In sharp contrast, state-of-the-art models need to be *retrained* for every new potential value (Kim et al., 2020).

**Zero-shot simulation of the DFR process**. Inferring the rate matrix and initial condition of a MJP process entails that one can also *sample from it*. Our FIM can thus be used as a *zero-shot generative model* for MJPs. However, to test the quality of said MJP realizations wrt. some target MJP, we need a distance between the two. Here we propose to use the Hellinger distance (Le Cam and Yang, 2000) to first estimate the divergence between a sequence of (local) histogram pairs, recorded at a given set of observation times, and then average the local estimates along time. Appendix F.1 empirically demonstrates that this pragmatically defined MJP distance is sensible.

Table 2 reports the time-averaged Hellinger distance between 1000 (ground-truth) DFR paths and 1000 paths sampled from (the MJPs inferred by) NeuralMJP and FIM. We repeat this calculation 100 times, for 1000 newly sampled paths from NeuralMJP and FIM, but the same 1000 target paths, to compute the mean values and error bars in the Table. The results show that the zero-shot DFR simulation obtained through FIM is on par with the NeuralMJP-based simulation, wrt. the ground truth.

### 4.2 Switching Ion Channel (IonCh): Zero-Shot Inference of Three-State MJP

In this section we study the conformational dynamics of the viral ion channel $\text{Kcv}_{\text{MT325}}$, which exhibits three metastable states (Gazzarrini et al., 2006). Specifically, we analyse the ion flow across the membrane as the system jumps between its metastable configurations. This ion flow was recorded at a frequency of 5kHz over one second. Figure 1 shows one snapshot of these recordings, which were made available to us via private communication (see the Acknowledgements). Our goal is to infer physical observables — like the stationary distribution and mean first-passage times — of the conformational dynamics, and to compare our findings against the SDiff model of Köhs et al. (2021) and NeuralMJP.

The recordings live in real space, which means that we first need to obtain a coarse-grained representation (CGR) from them, before we can apply FIM. Here we consider two CGRs: the CGR inferred

| Dataset | NEURALMJP | FIM |
|---|---|---|
| DFR | 0.30(0.06) | 0.27(0.06) |
| IONCH | 0.48(0.02) | **0.41(0.02)** |
| ADP | 1.38(0.52) | 1.39(0.47) |
| PFOLD | 0.015(0.015) | 0.014(0.014) |

|  | BOTTOM | MIDDLE | TOP |
|---|---|---|---|
| SDIFF | 0.17961 | 0.14987 | 0.67052 |
| NEURALMJP | 0.17672 | 0.09472 | 0.72856 |
| FIM-NMJP | 0.18224 | 0.10156 | 0.71621 |
| FIM-GMM | 0.19330 | 0.08124 | 0.72546 |

Table 2: Time-averaged Hellinger distances between empirical processes and samples from either NeuralMJP or FIM [in a 1e-2 scale] (lower is better). Mean and std. are computed from a set of 100 histograms

Table 3: Stationary distribution inferred from the switching ion channel experiment. FIM-NMJP and FIM-GMM correspond to our inference from different coarse-grained representations. The results agree well.

by NeuralMJP and a naive CGR obtained with a Gaussian Mixture Model (GMM). Given that we only have 5000 observations available, we make use of a single FIM evaluation with context number $c(50, 100)$. We infer two FIM rate matrices, one per each CGR, which we label as FIM-NMJP and FIM-GMM.

Table 3 contains the inferred stationary distributions from all models and evidences that a single FIM evaluation is enough to unveil the long-time asymptotics of the process. Similarly, Table 15 in the Appendix, which contains the inferred mean-first passage times, demonstrates that FIM makes the same inference about the short-term dynamics of the process as do SDiff and NeuralMJP. See Appendix F for additional results.

**Zero-shot simulation of switching ion channel process**. Just as we did with the DFR process, we can use FIM to simulate the switching ion channel process in coarse-grained space. Since only paths on the same CG space can be compared, we evaluate NeuralMJP against FIM-NMJP. To construct the target distribution, we leverage another 30 seconds of measurements, which amount to 150K observations that have not been seen by any of the models. The results in Table 2 indicate that our zero-shot simulations is statistically closer to the ground-truth process than the NeuralMJP simulation.

### 4.3 Alanine Dipeptide (ADP): Zero-Shot Inference of Six-State MJP

Alanine dipeptide is 22-atom molecule widely used as benchmark in molecular dynamics simulation studies. Its popularity stems from the fact that the heavy-atom dynamics, which jumps between six metastable states, can be fully described in terms of the dihedral (torsional) angles $\psi$ and $\phi$ (see *e.g.* Mironov et al. (2019) for details).

We examine an all-atom ADP simulation of 1 microsecond, which was made available to us via private communication (see the Acknowledgements below), and compare against both, the VampNets model of Mardt et al. (2017) and NeuralMJP. The data consists of the values taken by the dihedral angles as time evolves and thus needs to be mapped onto some coarse-grained space. We again make use of NeuralMJP to obtain a CGR. We then use FIM with context number $c(300, 100)$ to process 32 100-point time windows of the simulation and compute an average rate matrix. Note that this is the optimal context number of our pretrained model. Table 4 (and Appendix F.2) confirms that, once again, FIM can infer the same physical properties from the ADP simulation as the baselines.

**Zero-shot simulation of the alanine dipeptide**. Simulations in coarse-grained space for molecular dynamics is a high-interest research direction (Husic et al., 2020). Here we demonstrate that FIM can be used to simulate the ADP process in zero-shot mode. Indeed, Table 2 reports the distance from both NeuralMJP and FIM to a target ADP process, computed from 200 paths with 100 observations each. Once more, FIM performs comparable to NeuralMJP.

### 4.4 Zero-Shot Inference of Two-State MJPs

Finally, we consider two additional systems that feature jumps between two metastable states: a simple protein folding model and a two-mode switching system. We invite the reader to check out Appendix F.5 and F.6 for the details. That being said, Table 8 reports the distance of both NeuralMJP and FIM wrt. the empirical protein folding process (PFold). The high variance indicates that the distance cannot resolve any difference between the processes given the available number of samples.

| | PROBABILITY PER STATE | | | | | | RELAXATION TIME SCALES (IN $ns$) | | | | |
|---|---|---|---|---|---|---|---|---|---|---|---|
| | I | II | III | IV | V | VI | | | | | |
| VAMPNETS | 0.30 | 0.24 | 0.20 | 0.15 | 0.11 | 0.01 | 0.008 | 0.009 | 0.055 | 0.065 | 1.920 |
| NEURALMJP | 0.30 | 0.31 | 0.23 | 0.10 | 0.05 | 0.01 | 0.009 | 0.009 | 0.043 | 0.069 | 0.774 |
| FIM | 0.28 | 0.28 | 0.24 | 0.07 | 0.10 | 0.03 | 0.008 | 0.009 | 0.079 | 0.118 | 0.611 |

Table 4: *Left*: stationary distribution of the ADP process. The states are ordered in such a way that the ADP conformations associated with a given state are comparable between the VampNets and NeuralMJP CGRs. *Right*: relaxation time scales to stationarity. FIM agrees well with both baselines.

## 5 Conclusions

In this work we introduced a novel methodology for zero-shot inference of Markov jump processes and its Foundation Inference Model (FIM). We empirically demonstrated that *one and the same* FIM can be used to estimate stationary distributions, relaxation times, mean first-passage times, time-dependent moments and thermodynamic quantities (*i.e.* the entropy production) from noisy and discretely observed MJPs, taking values in state spaces of different dimensionalities, *all in zero-shot mode*. To the best of our knowledge, FIM is also the first zero-shot generative model for MJPs.

*Future work* shall involve extending our methodology to Birth and Death processes, as well as considering more complex (prior) transition rate distributions. See our discussion on Limitations in the next section, for details.

## 6 Limitations

The main limitations of our methodology clearly involve our synthetic distribution. Evaluating FIM on empirical datasets whose distribution significantly deviates from our synthetic distribution will, inevitably, yield poor estimates. Consider Figure 4 (right), for example. The performance of FIM quickly deteriorates for $V \geq 3$, for which the ratio between the largest and smallest rates gets larger than about three orders of magnitude. These cases are unlikely under our prior Beta distributions, and hence effectively lie outside of our synthetic distribution.

More generally, the MJP dynamics underlying phenomena that feature long-lived, metastable states, ultimately depends on the shape of the energy landscape characterizing the set $\mathcal{X}$, inasmuch as the transition rates between metastable states $i$ and $j$ ($f_{ij}$ in our notation) are characterized by *the depth of the energy traps* (that is, the height of the barrier between them).

In equations, we write

$$f_{ij} = \exp\left(\frac{-E_j}{T}\right),\tag{8}$$

where $E_j$ is the $j$th trap depth, and $T$ is the temperature of the system. Therefore, the distribution over energy traps determines the distribution over transition rates.

Just to give an example, if we studied systems with exponentially distributed energy traps — as *e.g.* in the classical Trap model of glassy systems of Bouchaud (1992) — we would immediately find $p(f) \propto Tf^{T-1}$. Transition rates sampled from such power-law distributions clearly lie outside our ensemble of Beta distributions, even if we use our rescaling trick. Future work shall explore training FIM on synthetic MJPs featuring power-law-distributed transition rates.

## Acknowledgements

This research has been funded by the Federal Ministry of Education and Research of Germany and the state of North-Rhine Westphalia as part of the Lamarr Institute for Machine Learning and Artificial Intelligence. Additionally, César Ojeda was supported by Deutsche Forschungsgemeinschaft (DFG) – Project-ID 318763901 – SFB1294.

We would like to thank Lukas Köhs for sharing the experimental ion channel data with us. The actual experiment was carried out by Kerri Kukovetz and Oliver Rauh while working in the lab of Gerhard

Thiel of TU Darmstadt. Similarly, we would like to thank Nick Charron and Cecilia Clementi, from the Theoretical and Computational Biophysics group of the Freie Universität Berlin, for sharing the all-atom alanine dipeptide simulation data with us. The simulation was carried out by Christoph Wehmeyer while working in the research group of Frank Noé of the Freie Universität Berlin.

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

# A  Background on MJPs

In this section we provide some brief background on MJPs and describe how physical quantities such as the stationary distributions, relaxation times and mean first passage times can be computed from the intensity matrix. Additionally, we mention how trajectories for MJPs can be sampled using the Gillespie algorithm.

## A.1  Background on Markov Jump Processes in Continuous Time

Markov jump processes are stochastic models used to describe systems that transition between states at random times. These processes are characterized by the Markov property where the future state depends only on the current state, not on the sequence of events that preceded it.

A continuous-time MJP $X(t)$ has right-continuous, piecewise-constant paths and takes values in a countable state space $\mathcal{X}$ over a time interval $[0, T]$. The instantaneous probability rate of transitioning from state $x'$ to $x$ is defined as

$$f(x|x', t) = \lim_{\Delta t \to 0} \frac{1}{\Delta t} p_{\text{MJP}}(x, t + \Delta t | x', t), \tag{9}$$

where $p_{\text{MJP}}(x, t | x', t')$ denotes the transition probability.

The evolution of the state probabilities $p_{\text{MJP}}(x, t)$ is governed by the master equation

$$\frac{dp_{\text{MJP}}(x, t)}{dt} = \sum_{x' \neq x} \Big( f(x|x') p_{\text{MJP}}(x', t) - f(x'|x) p_{\text{MJP}}(x, t) \Big). \tag{10}$$

For homogeneous MJPs with time-independent transition rates, the master equation in matrix form is

$$\frac{dp_{\text{MJP}}(x, t)}{dt}(t) = \mathbf{p}_{\text{MJP}}(t) \cdot \mathbf{F}, \tag{11}$$

with the solution given by the matrix exponential

$$\mathbf{p}_{\text{MJP}}(t) = \mathbf{p}_{\text{MJP}}(0) \cdot \exp(\mathbf{F}t). \tag{12}$$

## A.2  Stationary Distribution

The stationary distribution $\mathbf{p}_{\text{MJP}}^*$ of a homogeneous MJP is a probability distribution over the state space $\mathcal{X}$ that satisfies the condition $\mathbf{p}_{\text{MJP}}^* \cdot \mathbf{F} = \mathbf{0}$. This implies that the stationary distribution is a left eigenvector of the rate matrix corresponding to the eigenvalue 0.

## A.3  Relaxation Times

The relaxation time of a homogeneous MJP is determined by its non-zero eigenvalues $\lambda_2, \lambda_3, \ldots, \lambda_{|\mathcal{X}|}$. These eigenvalues define the time scales of the process: $|\text{Re}(\lambda_2)|^{-1}, |\text{Re}(\lambda_3)|^{-1}, \ldots, |\text{Re}(\lambda_{|\mathcal{X}|})|^{-1}$. These time scales are indicative of the exponential rates of decay toward the stationary distribution. The relaxation time, which is the longest of these time scales, dominates the long-term convergence behavior. If the eigenvalue corresponding to the relaxation time has a non-zero imaginary part, then this means that the system does not converge into a fixed stationary distribution but that it instead ends in a periodic oscillation.

## A.4  Mean First-Passage Times (MFPT)

For an MJP starting in a state $i \in \mathcal{X}$, the first-passage time to another state $j \in \mathcal{X}$ is defined as the earliest time $t$ at which the MJP reaches state $j$, given it started in state $i$. The mean first-passage time (MFPT) $\tau_{ij}$ is the expected value of this time. For a finite state, time-homogeneous MJP, the MFPTs can be determined by solving a series of linear equations for each state $j$, distinct from $i$, with the initial condition that $\tau_{ii} = 0$

$$\begin{cases} \tau_{ii} = 0 \\ 1 + \sum_k \mathbf{F}_{ik} \tau_{kj} = 0, & j \neq i \end{cases} \tag{13}$$

### A.5 The Gillespie Algorithm for Continuous-Time Markov Jump Processes

The Gillespie algorithm (Gillespie, 1977) is a stochastic simulation algorithm used to generate trajectories of Markov jump processes in continuous time. The algorithm proceeds as follows:

---

**Algorithm 1** Gillespie Algorithm for Markov Jump Processes

---

1: INPUT: The intensity matrix $\mathbf{F}$, the initial state distribution $\pi_0$, the starting time $t_0$ and the end time $t_{\text{end}}$
2: Initialize the time $t$ to the starting time $t_0$
3: Initialize the system's state $s$ to an initial state $s_0 \sim \pi_0$
4: While $t < t_{\text{end}}$ do
5:     Calculate the intensity $\lambda = -1/\mathbf{F}_{ss}$ from state $s$
6:     Sample the time $\tau$ to the next event from an exponential distribution with rate $\lambda$
7:     Update the time $t \leftarrow t + \tau$
8:     If $t \geq t_{\text{end}}$ then exit loop
9:     Calculate transition probabilities $p = -\mathbf{F}_{sj}/\mathbf{F}_{ss}$ for each possible next state $j$
10:    Set $p_s$ to zero because we allow for no self jumps
11:    Sample the next state $s'$ from the distribution defined by $p$
12:    Update the system's state $s \leftarrow s'$
13:    Record the state $s$ and time $t$
14: End while
15: OUTPUT: The trajectory of states and times

---

## B Synthetic Dataset Generation: Statistics and other Details

This section is a continuation of section 3.1 and provides more details on the generation of our synthetic training dataset. Additionally, we provide some statistics about the dataset distribution.

### B.1 Prior Distributions and their Implementation

In this subsection we give additional details about our data generation mechanism.

**Distribution over rate matrices**. Our data generation procedure starts by sampling the entries $f_{ij}$ of the intensity matrix from the following beta distributions

$$p(f_{ij}|\rho_f) = \text{Beta}(\rho_f = (\alpha, \beta)), \ \text{with} \ p(\alpha) = \text{Uniform}(\{1, 2\})$$
$$\text{and} \ p(\beta) = \text{Uniform}(\{1, 3, 5, 10\}). \quad (14)$$

Both these discrete uniform distribution define the prior $p(\rho_f) = p(\alpha)p(\beta)$.

The choices for $\alpha$ and $\beta$ were made heuristically, to obtain reasonable (*i.e.* varied) distributions over the number of jumps (see *e.g.* Figure 5). We remark that we fixed this set of training distributions *before evaluating the model on the evaluation sets*, in order to prevent us from introducing *unwanted* biases into the distribution hyperparameters by optimizing on the evaluation set.

Next we define the prior over the adjacency matrix as

$$p(\mathbf{A}) = \frac{1}{2}\delta(\mathbf{A} - \mathbf{J}) + \frac{1}{2}p_{\text{Erdös-Rényi}}(\mathbf{A}, p = 0.5) \quad (15)$$

where $\delta(\cdot)$ labels the Dirac delta distribution and $\mathbf{J}$ denotes the matrix for which all off-diagonal entries are 1 and the diagonal ones are 0. Furthermore $p_{\text{Erdös-Rényi}}$ labels the Erdös-Rényi model (Erdös and Rényi, 1959), for which each link is defined via an independent Bernoulli variable, with some fixed, global probability $p$, here set to $\frac{1}{2}$. Equation 15 indicates that (in average) 50 percent of our state networks are fully connected, whether the other 50 percent are not.

Our motivation for this prior is that it often happens in real world processes that the intensity matrices are not fully connected. Let us remark, however, that we only accept the Erdös-Rényi sample if the corresponding graph is *connected* — that is, if the system cannot get stuck into a single state. Both these distributions implicitly define $p_{\text{rates}}(\mathbf{F}|\mathbf{A}, \rho_f)$, for $F_{ij} = a_{ij}f_{ij}$.

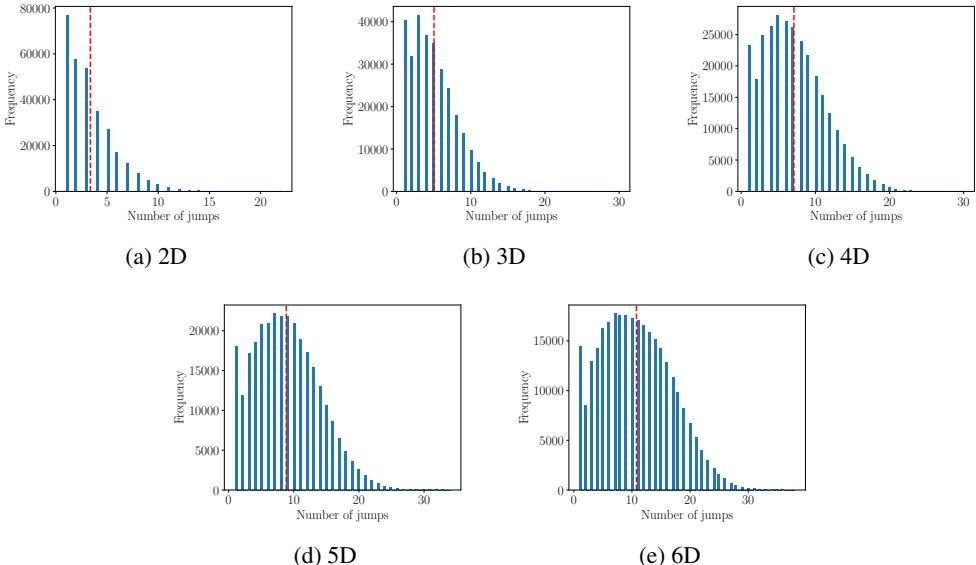

Figure 5: Distributions of the number of jumps per trajectory. We used the same distributions as the training set and sampled up to time 10. The figures are based on 1000 processes with 300 paths per process.

*Remark on generalization beyond prior rate distribution.* We remark that while all entries of the intensity matrix seen during training lie on the interval $[0, 1]$, the model can still predict intensities outside this interval. We empirically demonstrated that this in indeed the case on the widely different target sets of the experimental section, in the main text. The reason behind this is that we normalize the maximum time among all input paths to be 1, and rescale the predicted intensities accordingly. Ultimately, what matters is the difference among the rates (and therefore among the observation times) within the target time series. Our approach for sampling intensity matrices resulted in a vast variety of different processes.

The distribution of the number of jumps per trajectory is shown in figure 5 and that of relaxation times is shown in figure 6.

**Distribution over initial conditions**. We choose half of our initial distributions in our synthetic ensemble to be *the stationary distribution of the MJP* $p_{\text{MJP}}^*$. The motivation for this is that it often happens that real life experiments produce very long observations of a system in equilibrium. The second half of our initial distributions $\boldsymbol{\pi}_0$ are randomly sampled from a Dirichlet distribution $\text{Dir}(\rho_0)$, where we heuristically choose $\rho_0 = 50$. In equations, we write

$$p(\boldsymbol{\pi}_0) = \frac{1}{2} p_{\text{MJP}}^*(\mathbf{F}) + \frac{1}{2} \text{Dir}(\rho_0 = 50). \tag{16}$$

**Distribution over observation grids**. In practice, the exact jump (*i.e.* transition) times are not known. We therefore first generate observations of the state of the system on a regular grid with a maximum of $L = 100$ points. We then randomly mask out some observations from this fixed regular grid, in order to make the model grid independent. Half of our (subsampled) observation grids are chosen to be *regular*, *i.e.* they are strided with strides $\in \{1, 2, 3, 4\}$. The other half are chosen to be *irregular*, through a Bernoulli filter (or mask) with $\rho_{\text{survival}} \in \{1/4, 1/2\}$ applied to the base ($L = 100$) grid.

**Distribution over noise process**. Because real world data is often noisy we also add noise to the labels. If a state observation is selected to be mislabeled, the new label is randomly chosen from a uniform distribution over all states. We investigate two different configurations in this project, one with 1% label noise ($\rho_x = 0.01$) and one with 10% label noise ($\rho_x = 0.1$).

**MJP simulation**. We sample the jumps between different states with an algorithm due to (Gillespie, 1977) (see A.5). We sample jumps between times 0 and 10 because almost all of our processes are in equilibrium by then (see figure 6).

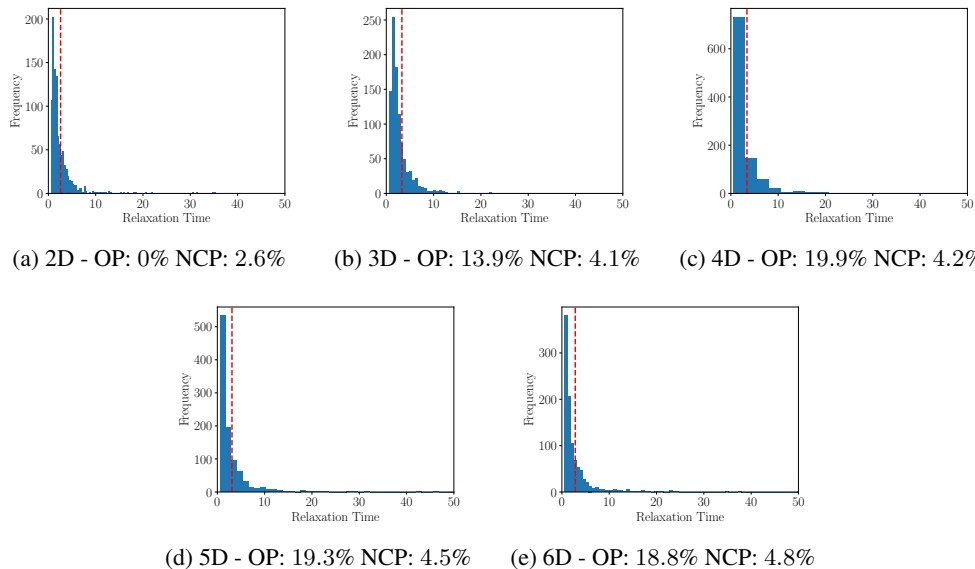

(a) 2D - OP: 0% NCP: 2.6%   (b) 3D - OP: 13.9% NCP: 4.1%   (c) 4D - OP: 19.9% NCP: 4.2%

(d) 5D - OP: 19.3% NCP: 4.5%   (e) 6D - OP: 18.8% NCP: 4.8%

Figure 6: Distributions of the relaxation times. We also report the percentage of processes that converge into an oscillating distribution (OP) and the percentage of processes that have a relaxation time which is larger than the maximum sampling time (NCP) of our training data (given by $t_{\text{end}} = 10$). The figures are based on 1000 processes.

**Training Dataset Size** The synthetic dataset on which our models were trained consists of 25k six-state processes, and 5k processes of 2-5 states, resulting in a total size of 45k processes. For each of these processes we sampled 300 paths.

**Distribution over the number of MJP paths** $p(K)$. While we generate the data with 300 paths per process, we want to ensure that the model is able to handle datasets with less than 300 paths. For this reason, we shuffle the training data at the beginning of every epoch and distribute it into batches with path counts $1, 11, 21, \ldots, 300$. We found that such a static selection of the path counts is better than a random selection, because a random selection leads to oscillating loss functions (because the model obviously gets a larger loss for samples with fewer paths), and thus training instabilities. Since we do not always select all paths per process but instead select a random subset of them, the data that the model processes changes during every epoch, which helps in reducing overfitting.

## C   How to use the Model: Inputs, Outputs and Rescalings

In this section, we give details about the inputs to and outputs of our pretrained recognition model. We also comment on the internal rescalings done by the model, in order to be able to infer MJPs from time series with observation times of any scale.

### C.1   Input

The model takes as input three parameters:

1. The `observation grids` (shape: [`num_paths` $K$, `grid_size` $L$]): The observation times $\{\tau_{k1}, \ldots, \tau_{kl}\}_{k=1}^{K}$, padded to the maximum length $L$.
2. The `observation values` (shape: [`num_paths` $K$, `grid_size` $L$]): The noisy observation values $\{x'_{k1}, \ldots, x'_{kl}\}_{k=1}^{K}$ padded to the maximum length $L$. Note that these values are integers lying on the discrete set $\{0, 1, \ldots, C-1\}$.
3. The `dimension` $c$: The (a priori known) dimension of the process as an integer between 2 and $C$. If this dimension is unknown, the model returns a $C \times C$ rate matrix whose rank might (approximately) be smaller than $C$, which indicates a hidden state-space of size smaller than $C$.

We recommend users to use the model only within its training range. That is, with up to a maximum of $K = 300$ paths, and grids up to a maximum of $L = 100$ points.

## C.2 Internal Rescaling

Internally, the model does the following:

1. It computes the maximum observation time:

$$\tau_{\max} = \max\{\tau_{k1}, \ldots, \tau_{kl}\}_{k=1}^{K}. \tag{17}$$

2. It normalizes the observation times between 0 and 1:

$$\{\tau_{k1}, \ldots, \tau_{kl}\}_{k=1}^{K} \leftarrow \{\tau_{k1}, \ldots, \tau_{kl}\}_{k=1}^{K}/\tau_{\max}. \tag{18}$$

3. It computes the inter-event times $\Delta\tau_{ki} = \tau_{k,i+1} - \tau_{ki}$, for $k = 1, \ldots, K$.

4. It transforms the observation values to one-hot-encodings.

5. It predicts the (normalized) off-diagonal elements of the intensity matrix and variance matrix as well as the initial distribution (here we are working with the maximum supported dimension, that is $C$).

6. It rescales back the estimates to the original time scale:

$$\text{intensity matrix } \hat{\mathbf{F}} \leftarrow \text{intensity matrix } \hat{\mathbf{F}}/\tau_{\max}, \tag{19}$$

$$\text{Var } \hat{\mathbf{F}} \leftarrow \text{Var } \hat{\mathbf{F}}/\tau_{\max}. \tag{20}$$

Note that, as we empirically demonstrated in the paper, this rescaling procedure allows us to work with real-world MJPs of arbitrary time scales. For example, the time scales for the switching ion channel dataset were more than 500 times smaller than the time scales in our training dataset.

## C.3 Support for varying State Space Sizes

We now elaborate on how the model can deal with processes whose state spaces have sizes $c < C$.

We arranged all the target rate matrices $\mathbf{F}$ within our training dataset, for MJPs with state spaces of size $c < C$, to be the leftmost block diagonal $c \times c$ matrix within a $C \times C$ matrix of zeros, so that the redundant matrix elements are always zero. As can be read from equation (6) of the main text, we train FIM to predict zeros for those redundant matrix elements.

In practice, however, our trained FIM does not exactly predict zeros for those redundant matrix elements. In our experiments, the user knows a priori the number of states $c$ of the hidden process, so we explicitly set the redundant matrix elements to zero, and only then compute the corrected diagonal (i.e. the normalization) of the output rate matrix.

We afterwards select the $c \times c$ entries of the predicted intensity matrix and variance matrix as well as the first $c$ entries of the predicted initial distribution.

We refer the reader to our library for additional details.

## C.4 Output

The output of the model consists of three parameters:

1. The intensity matrix $\hat{\mathbf{F}}$ (shape: $[c, c]$).

2. The variance matrix Var $\hat{\mathbf{F}}$ (shape: $[c,c]$).

3. The initial distribution $\pi_0$ (shape $[c]$).

# D   Model Architecture and Experimental Setup

In this section we provide more details about the architecture of our models and the hyperparameters.

## D.1   Model Architecture

**Path encoder** $\psi_1$. We evaluated two different approaches for the path encoder $\psi_1$. The first approach utilizes a bidirectional LSTM (Hochreiter and Schmidhuber, 1997) as $\psi_1$, while the second approach employs a transformer (Vaswani et al., 2017) for $\psi_1$. The time series embeddings are denoted by $h_{k\theta}$ (see Equation 4). The input to the encoder $\psi_1$ is $(\mathbf{x}'_{k1}, \boldsymbol{\tau}_{k1}, \ldots, \mathbf{x}'_{kl}, \boldsymbol{\tau}_{kl})$, where $\boldsymbol{\tau}_{kl} = [\tau_{kl}, \delta_{kl}]$, $\delta_{kl} = \tau_{kl} - \tau_{(k-1)l}$, and $\mathbf{x}_{kl} \in \{0, 1\}^C$ is the one-hot encoding of the system's state.

**Path attention network** $\Omega_1$. We tested two approaches. The first approach uses classical self-attention Vaswani et al. (2017) and selects the last embedding. For the second approach we used an approach we denote as *learnable query attention* which is equivalent to classical multi-head attention with the exception that we do not compute the query based on the input, but instead make it a learnable parameter, i.e.,

$$\text{MultiHead}(Q, K, V) = \text{Concat}(\text{head}_1, \ldots, \text{head}_h), \tag{21}$$

$$\text{head}_i = \text{Attention}(Q_i, H_{1:K}W_i^K, H_{1:K}W_i^V), \tag{22}$$

where $H_{1:K} \in \mathbb{R}^{K \times d_{model}}$ denotes a concatenation of $h_1, \ldots, h_K$, $W_i^K, W_i^V \in \mathbb{R}^{d_{model} \times d_k}$ and $Q_i \in \mathbb{R}^{q \times d_k}$ is the learnable query matrix. The output dimension of the learnable query attention is therefore independent of the number of input tokens.

## D.2   Experimental Setup

**Hyperparameter tuning:** Hyperparameters were tuned using a grid search method. The optimizer utilized was AdamW (Loshchilov and Hutter, 2017), with a learning rate and weight decay both set at $1e^{-4}$. A batch size of 128 was used. During the grid search, we experimented with the hidden size of the *path encoder* ([64, 128, 256, 512]), the hidden size of the *path attention network* ([128, 256]), and various MLP architectures for $\phi_1, \phi_2$, and $\phi_3$ ([[32, 32], [128, 128]]).

**Training procedure:** All models were trained on two A100 80Gb GPUs for approximately 500 epochs or approximately 2.5 days on average per model. Early stopping was employed as the stopping criterion. The models were trained by maximizing the likelihood.

**Final model parameters**: The final models (FIM-MJP 1% Noise and FIM-MJP 10% Noise) have the following hyperparameters: *Path encoder* - hidden_size($\psi_1$) $= 256$ (the final models used a BiLSTM); *Path attention network* - $\Omega_1$: $q = 16$, $d_k = 128$ (the final models used the learnable query approach); $\phi_1, \phi_2, \phi_3 = [128, 128]$.

**Pretrained models**: Our pretrained models are also available online[5].

# E   Ablation Studies

In this section, we study the performance of the models with different architectures. Additionally, we study the behavior of the performance of the models with respect to varying numbers of states and varying number of paths.

## E.1   General Remarks about the Error Bars and Context Number

If the evaluation set is larger than the optimal context number $c(K_{max}, l_{max})$, we split the evaluation set into batches and give these to the model independently (because the model does not work well to give the model more paths than during training, see table 8). Afterwards, we compute the mean of the predictions among the batches and report the mean RMSE of the intensity entries (if the ground-truth is available). This makes it easier to compare our model against previous works which have also used the full dataset to make predictions. Interestingly, we find that the RMSE of this averaged prediction

---

[5] https://github.com/cvejoski/OpenFIM

is often significantly better than the mean RMSE among the batches. For example for the DFR dataset the RMSE of the averaged prediction is 0.0617, while the average RMSE of the batches is 0.122. If the dataset has been split into multiple batches, we report the RMSE together with the standard deviation of the RMSE among the batches. The reported confidence is the mean predicted variance of the model (recall that we are using Gaussian log-likelihood during training).

## E.2 Performance of the Model by varying its Architecture

The ablation study presented in Table 5 evaluates the impact of different model features on the performance by comparing various combinations of architectures and attention mechanisms with varying numbers of paths, and their corresponding RMSE values. The study examines models using a BiLSTM or Transformer, with and without self-attention and learnable query attention, across 1, 100, and 300 paths. The results indicate that increasing the number of paths consistently reduces RMSE (see section E.4 for more details), demonstrating the benefit of considering more paths during training. Specifically, using a BiLSTM with learnable query attention achieves an RMSE of $0.193 \pm 0.031$ with a single path, significantly improving to $0.048 \pm 0.011$ with 100 paths, and further to $0.0457 \pm 0.0$ with 300 paths. Similarly, a Transformer with learnable query attention shows an RMSE of $0.196 \pm 0.031$ for a single path, $0.049 \pm 0.011$ for 100 paths, and $0.0458 \pm 0.0$ for 300 paths. The inclusion of self-attention in the Transformer models slightly improves performance, with the best RMSE of $0.0459 \pm 0.0$ achieved when both self-attention and learnable query attention are used with 300 paths. In this case since many of the processes contain one path it is beneficial to use the learnable query attention over the standard self-attention mechanism.

| # Paths | BiLSTM | Transformer | Self Attention | Learnable Query Attention | RMSE |
|---------|--------|-------------|----------------|---------------------------|------|
| 1 | ✓ | | | ✓ | $0.193 \pm 0.031$ |
| 1 | ✓ | | ✓ | | $0.196 \pm 0.031$ |
| 1 | | ✓ | | ✓ | $0.197 \pm 0.015$ |
| 100 | ✓ | | | ✓ | $0.048 \pm 0.011$ |
| 100 | ✓ | | ✓ | | $0.049 \pm 0.011$ |
| 100 | | ✓ | | ✓ | $0.054 \pm 0.012$ |
| 300 | ✓ | | | ✓ | $0.0457 \pm 0.0$ |
| 300 | ✓ | | ✓ | | $0.0458 \pm 0.0$ |
| 300 | | ✓ | | ✓ | $0.0459 \pm 0.0$ |

Table 5: Comparison of model features with different number of paths and their RMSE. This table presents an ablation study comparing the performance of models using BiLSTM and Transformer architectures, with and without self-attention and learnable query attention, across different numbers of paths (1, 100, and 300). The performance is measured by the Root Mean Square Error (RMSE), with lower values indicating better model accuracy. The study highlights that both the architectural choices and the number of paths significantly impact model performance, with the best results achieved using a combination of attention mechanisms and a higher number of paths.

Figure 7 presents a series of line plots illustrating the impact of different hyperparameter settings on the RMSE of the model. The first subplot shows the RMSE as a function of the hidden size of the $\psi_1$ path encoder, with hidden sizes 64, 128, 256, and 512. The RMSE increases as the hidden size increases, with the lowest RMSE observed at a hidden size of 256. The second subplot displays the RMSE as a function of the architecture size of $\phi_1$, comparing two architectures: [2x32] and [2x128]. The RMSE decreases as the architecture size increases, indicating better performance with a larger architecture size for $\phi_1$. The third subplot examines the RMSE based on the architecture size of $\phi_2$, with two architectures tested: [2x32] and [2x128]. There is no significant difference in RMSE between the two sizes, suggesting that the choice of architecture size for $\phi_2$ does not markedly affect model performance. The fourth subplot investigates the RMSE as a function of the hidden size of the $\Omega_1$ component, with hidden sizes 128 and 256 tested, and results shown for different $\psi_1$ hidden sizes (64, 128, 256, and 512). The RMSE remains relatively stable across different hidden sizes of $\Omega_1$, with slight variations observed depending on the hidden size of $\psi_1$. Overall, the plots highlight that some components, such as $\psi_1$ and $\phi_1$, are more sensitive to changes in hyperparameters, emphasizing the importance of selecting appropriate hyperparameters to optimize model performance.

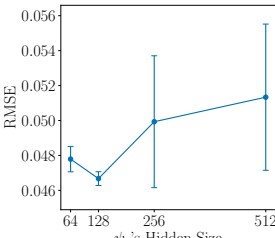 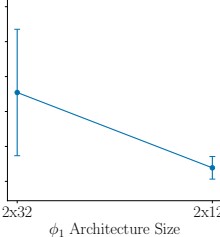 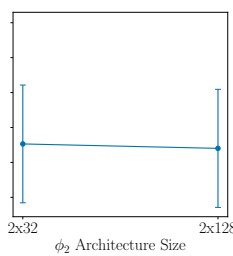 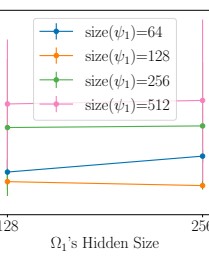

Figure 7: Impact of Hyperparameters on RMSE. The figure shows four line plots illustrating the effect of hyperparameters on model RMSE. The first plot shows RMSE increases with larger $\psi_1$ hidden sizes, being lowest at 256. The second plot indicates lower RMSE with a larger $\phi_1$ architecture size ([2x128]). The third plot shows minimal RMSE impact from $\phi_2$ architecture size. The fourth plot shows RMSE stability across different $\Omega_1$ hidden sizes, with slight variations based on $\psi_1$. This highlights the importance of tuning $\psi_1$ and $\phi_1$ for optimal performance.

|  | 1% Noise Data | 10% Noise Data |
|---|---|---|
| FIM-MJP 1% | 0.046 | 0.199 |
| FIM-MJP 10% | 0.096 | 0.087 |

Table 6: Performance of FIM-MJP 1% and FIM-MJP 10% on synthetic datasets with different noise levels. We use a weighted average among the datasets with different numbers of states to compute a final RMSE.

Table 6 compares the performance of two models, FIM-MJP 1% and FIM-MJP 10%, on synthetic datasets with noise levels of 1% and 10%, measured in terms of RMSE. For datasets with 1% noise, the FIM-MJP 1% model achieves an RMSE of 0.046, indicating good performance, but its RMSE increases significantly to 0.199 on 10% noise data, showing decreased performance with higher noise. Conversely, the FIM-MJP 10% model, trained with 10% noise data, has an RMSE of 0.096 on 1% noise data, higher than the FIM-MJP 1% model on the same data, but achieves a lower RMSE of 0.087 on 10% noise data, demonstrating better performance under high noise conditions. This indicates that the FIM-MJP 10% model is more robust to noise, maintaining consistent performance across varying noise levels, while the FIM-MJP 1% model excels in low noise environments but struggles with higher noise. The results highlight the importance of training with appropriate noise levels to ensure robust model performance across different noise conditions.

### E.3 Performance of the Model with varying Number of States

We compare the performance of our models on processes with varying number of states. Note that our model always outputs a $6 \times 6$ dimensional intensity matrix. However, in these experiments we only use the rows and columns that correspond to the lower-dimensional process. This improves the comparability between different dimensions as lower-dimensional processes obviously have many zero-entries in their intensity matrix which would make it easier for the model to achieve a good RMSE score.

It can be seen in Table 7 that the multi-state-model performs well among all different dimensions. As expected, lower-dimensional processes seem to be easier for the model. Additionally, Table 7 shows the performance of a model which has only been trained on six-state processes. The performance of this native six-state-model for six number of states is very similar to the multi-state-model which shows that having more states during training does not reduce the single-state performance. As expected, the performance of the six-state model on processes with lower numbers of states is significantly worse, but still better than random.

### E.4 Performance of the Model with varying Number of Paths during Evaluation

One of the advantages of our model architecture is that it can handle arbitrary number of paths. We therefore use our model that was trained on at maximum 300 paths and assess its performance

| # States | Multi-State RMSE | Multi-State Confidence | 6-State RMSE | 6-State Confidence |
|---|---|---|---|---|
| 2 | 0.026 | 0.028 | 0.129 | 0.056 |
| 3 | 0.037 | 0.030 | 0.113 | 0.049 |
| 4 | 0.046 | 0.037 | 0.087 | 0.046 |
| 5 | 0.054 | 0.040 | 0.066 | 0.041 |
| 6 | 0.059 | 0.044 | 0.059 | 0.044 |

Table 7: Performance of the multi-state and six-state models (which has only been trained on processes with six states) on synthetic test sets with varying number of states

with varying number of paths during evaluation. The results are presented in Table 8. When being inside the training range, the performance and the confidence of the model goes down as the model is given fewer paths per evaluation, which is to be expected. Interestingly, the performance of the learnable-query (LQ) model peaks at 500 paths instead of at 300, which was the maximum training range. One possible explanation for this might be that we are still close enough to the training range while being able to use the full data (note that the dataset contains 5000 paths which is not divisible by 300, so we have to leave some of the data out). Going too far beyond the training range does however not work well, for example processing all 5000 paths at once leads to very poor performance, although the model (falsely) become very confident. Another insight from this experiment is that the self-attention (SA) architecture behaves significantly worse when going beyond the maximum number of paths that was seen during training. This is another reason why we chose the (LQ) architecture over the (SA) architecture for the final version of our model.

| #Paths during Evaluation | RMSE (LQ) | Confidence (LQ) | RMSE (SA) | Confidence (SA) |
|---|---|---|---|---|
| 1 | $0.548 \pm 0.067$ | 0.838 | $0.579 \pm 0.074$ | 0.898 |
| 30 | $0.074 \pm 0.081$ | 0.263 | $0.075 \pm 0.070$ | 0.264 |
| 100 | $0.061 \pm 0.039$ | 0.143 | $0.060 \pm 0.035$ | 0.142 |
| 300 | $0.056 \pm 0.023$ | 0.089 | $0.059 \pm 0.024$ | 0.085 |
| 500 | $0.053 \pm 0.014$ | 0.069 | $0.074 \pm 0.021$ | 0.061 |
| 1000 | $0.067 \pm 0.012$ | 0.037 | $0.229 \pm 0.025$ | 0.029 |
| 5000 | $0.818 \pm 0.000$ | 0.000 | $2.135 \pm 0.000$ | 0.000 |

Table 8: Performance of FIM-MJP 1% given varying number of paths during the evaluation on the DFR dataset with regular grid. (LQ) denotes learnable-query-attention (see section D.1), (SA) denotes self-attention.

## F Additional Results

This section contains more of our results which did not fit into the main text. We begin this section by providing more details on the Hellinger distance which we used as a metric to assess the performance of our models. Afterwards, we provide more results and background on the ADP, ion channel and DFR datasets. Additionally, we introduce two two-state MJPs, given by the protein folding datasets (F.5) and the two-mode switching system (F.6), which we use to evaluate our models and to compare it against previous works.

### F.1 Hellinger Distance

Real-world empirical datasets of MJPs provide no knowledge of a ground truth solution. For this reason we present a new metric that can be used to compare the performance of the inference of various models based on only the empirical data. Our metric of choice is the Hellinger distance which is a measure of the dissimilarity between two probability distributions. Given two discrete probability distributions $P = (p_1, \ldots, p_k)$ and $Q = (q_1, \ldots, q_k)$, the Hellinger distance is defined as

$$H(P, Q) = \frac{1}{\sqrt{2}} \sqrt{\sum_{i=1}^{k} (\sqrt{p_i} - \sqrt{q_i})^2} . \tag{23}$$

For our empirical cases, the class probabilities of the discrete probability distributions are not known explicitly. We therefore approximate them by using the empirical distributions, given by the (normalized) histograms of the observed states at the observation grids.

We test this approach on the DFR process by first sampling a specified number of paths for the potential $V = 1$ using the Gillespie algorithm, which we then consider as the target distribution. Counting states among the different paths then yields histograms of the states for every time step. We repeat the same procedure for different choices of $V$. Afterwards we compute the Hellinger distance between the newly sampled histogram and the target distribution for every time step. Figure 8 shows that the distance indeed goes down as we approach the target distribution, which provides heuristic evidence of the effectiveness of our metric. The Hellinger distances for various models are shown in Table 2 and Table 9.

As one can see, FIM-MJP performs as well (and sometimes better) as the current state-of-the-art model NeuralMJP.

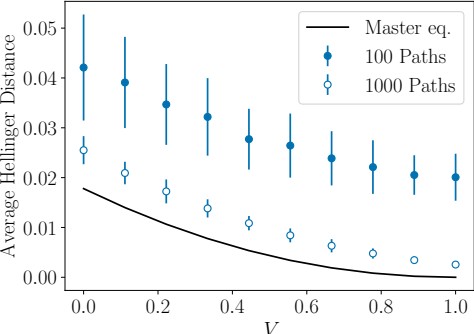

Figure 8: Time-Average Hellinger distance for varying potentials on the DFR. The plot shows the Hellinger distance to a target dataset that was sampled from a DFR with $V = 1$ on a grid of 50 points between 0 and 2.5. The means and standard deviations were computed by sampling 100 histograms per dataset. As expected, the distance decreases as the voltage gets closer to the voltage of the target dataset. We also remark that the scale of the distances gets smaller as one takes more paths into account and converge to the distance of the solutions of the master equation.

| Dataset | NeuralMJP | FIM-MJP 1% Noise | FIM-MJP 10% Noise |
|---|---|---|---|
| ADP | $1.38 \pm 0.52$ | $1.39 \pm 0.47$ | $1.35 \pm 0.42$ |
| Ion Channel | $0.48 \pm 0.02$ | $0.41 \pm 0.02$ | $1.78 \pm 0.03$ |
| Protein Folding | $0.015 \pm 0.015$ | $0.014 \pm 0.014$ | $0.024 \pm 0.026$ |
| DFR | $0.30 \pm 0.06$ | $0.27 \pm 0.06$ | $0.28 \pm 0.06$ |

Table 9: Comparison of the time-average Hellinger distances for various models. We used the same labels as NeuralMJP to make the results comparable. The errors are the standard deviation among 100 sampled histograms. The target datasets contain 200 paths for ADP, 1500 paths for Ion Channel, 2000 paths for Protein Folding and 1000 paths for the DFR. The distances are reported in a scale 1e-2. We remark that the high variance of the distances on the Protein Folding dataset is caused by the models performing basically perfect predictions, which causes the oscillations to be noise. We verified this claim by confirming that the distances of the predictions of the models are as small as the distance of the target dataset to additional simulated data.

## F.2 Alanine Dipeptide

We use the dataset of Husic et al. (2020), which models the conformal dynamics of ADP, for evaluating our model. This dataset was provided to us via private communication. The dataset consists of 9800 paths on grids of size 100 and has the sines and cosines of the Ramachandran angles as features:

$\sin\psi$, $\cos\psi$, $\sin\phi$ and $\cos\phi$. We use KMeans to classify the data into states. The reason why we did not choose GMM as for the other datasets is that we could initialize KMeans with hand-selected values to try to achieve a similar classification like those learned by NeuralMJP (Seifner and Sánchez, 2023), see Figure 9. Still, the classification is very different and thus also leads to very different results (see Table 10). We use 9600 paths to evaluate our models. Our results are shown in Table 10. Table 11 reports the stationary distributions and compares them to previous works, while Table 12 reports the ordered time scales.

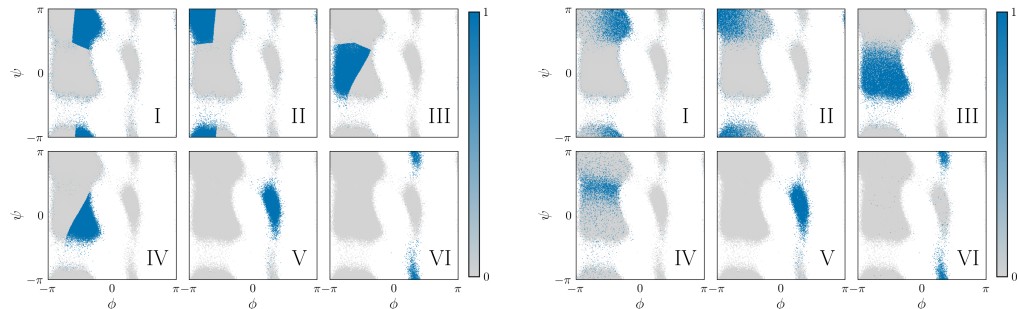

Figure 9: Comparison of the classifications between KMeans (left) and NeuralMJP (right).

| Model | Intensity Matrix | | | | | |
|---|---|---|---|---|---|---|
| NeuralMJP | $\begin{bmatrix} -61.32 & 53.15 & 0.19 & 7.89 & 0.06 & 0.02 \\ 47.29 & -59.37 & 0.05 & 11.97 & 0.04 & 0.01 \\ 0.28 & 0.13 & -17.28 & 16.81 & 0.02 & 0.04 \\ 35.48 & 26.94 & 40.93 & -103.61 & 0.25 & 0.01 \\ 0.16 & 0.22 & 0.31 & 0.2 & -3.86 & 2.96 \\ 1.13 & 1.73 & 0.46 & 0.66 & 18.78 & -22.76 \end{bmatrix}$ | | | | | |
| FIM-MJP 1% Noise (NeuralMJP Labels) | $\begin{bmatrix} -59.35 \pm 2.11 & 48.72 \pm 1.90 & 0.33 \pm 0.08 & 10.14 \pm 1.47 & 0.09 \pm 0.08 & 0.07 \pm 0.07 \\ 50.54 \pm 3.25 & -57.62 \pm 2.99 & 0.44 \pm 0.04 & 6.44 \pm 1.18 & 0.09 \pm 0.09 & 0.10 \pm 0.10 \\ 0.40 \pm 0.08 & 0.50 \pm 0.10 & -14.29 \pm 1.57 & 13.16 \pm 1.31 & 0.17 \pm 0.17 & 0.07 \pm 0.06 \\ 38.31 \pm 4.63 & 33.71 \pm 4.97 & 49.14 \pm 4.80 & -121.66 \pm 7.36 & 0.21 \pm 0.19 & 0.30 \pm 0.32 \\ 0.25 \pm 0.27 & 0.43 \pm 0.60 & 0.20 \pm 0.24 & 0.30 \pm 0.33 & -2.40 \pm 3.06 & 1.23 \pm 1.69 \\ 0.44 \pm 0.45 & 1.12 \pm 1.55 & 0.48 \pm 0.42 & 0.68 \pm 0.99 & 4.79 \pm 5.91 & -7.52 \pm 8.64 \end{bmatrix}$ | | | | | |
| FIM-MJP 10% Noise (NeuralMJP Labels) | $\begin{bmatrix} -49.35 \pm 4.58 & 40.51 \pm 3.82 & 0.3 \pm 0.1 & 7.96 \pm 1.69 & 0.35 \pm 0.15 & 0.22 \pm 0.11 \\ 39.99 \pm 6.65 & -46.82 \pm 6.37 & 0.3 \pm 0.1 & 5.99 \pm 1.14 & 0.27 \pm 0.07 & 0.27 \pm 0.08 \\ 0.27 \pm 0.04 & 0.44 \pm 0.1 & -13.05 \pm 1.66 & 11.35 \pm 1.81 & 0.32 \pm 0.07 & 0.68 \pm 0.27 \\ 39.18 \pm 5.42 & 28.24 \pm 4.14 & 58.86 \pm 8.72 & -129.02 \pm 10.51 & 1.1 \pm 0.19 & 1.64 \pm 0.58 \\ 9.61 \pm 7.02 & 9.32 \pm 6.83 & 5.53 \pm 3.97 & 4.36 \pm 3.09 & -43.11 \pm 29.51 & 14.3 \pm 9.01 \\ 2.49 \pm 1.12 & 5.8 \pm 2.25 & 8.82 \pm 4.95 & 6.72 \pm 2.32 & 11.5 \pm 5.67 & -35.32 \pm 5.99 \end{bmatrix}$ | | | | | |
| FIM-MJP 1% Noise (KMeans Labels) | $\begin{bmatrix} -175.42 \pm 8.87 & 172.65 \pm 8.73 & 1.84 \pm 0.69 & 0.48 \pm 0.12 & 0.22 \pm 0.18 & 0.23 \pm 0.24 \\ 157.16 \pm 13.99 & -165.37 \pm 13.64 & 6.67 \pm 1.78 & 1.17 \pm 0.24 & 0.22 \pm 0.16 & 0.14 \pm 0.14 \\ 22.26 \pm 3.88 & 9.84 \pm 3.10 & -375.78 \pm 20.96 & 342.13 \pm 19.80 & 0.71 \pm 0.67 & 0.84 \pm 0.65 \\ 0.93 \pm 0.15 & 1.37 \pm 0.16 & 305.86 \pm 20.47 & -308.48 \pm 20.30 & 0.25 \pm 0.19 & 0.07 \pm 0.09 \\ 0.81 \pm 1.34 & 0.35 \pm 0.39 & 0.28 \pm 0.29 & 0.25 \pm 0.27 & -2.30 \pm 2.52 & 0.61 \pm 0.82 \\ 0.28 \pm 0.33 & 0.89 \pm 1.14 & 0.28 \pm 0.38 & 0.18 \pm 0.23 & 4.81 \pm 7.13 & -6.44 \pm 9.08 \end{bmatrix}$ | | | | | |
| FIM-MJP 10% Noise (KMeans Labels) | $\begin{bmatrix} -94.75 \pm 15.46 & 91.38 \pm 16.21 & 1.91 \pm 0.76 & 0.84 \pm 0.15 & 0.32 \pm 0.09 & 0.29 \pm 0.10 \\ 184.85 \pm 20.63 & -190.00 \pm 19.41 & 1.98 \pm 0.49 & 0.49 \pm 0.23 & 0.84 \pm 0.32 & 1.83 \pm 0.93 \\ 5.93 \pm 1.57 & 13.71 \pm 2.48 & -266.49 \pm 18.43 & 241.54 \pm 17.99 & 0.85 \pm 0.18 & 4.48 \pm 0.52 \\ 1.44 \pm 0.74 & 0.91 \pm 0.35 & 188.88 \pm 31.10 & -193.76 \pm 29.77 & 1.29 \pm 0.30 & 1.22 \pm 0.31 \\ 3.45 \pm 1.82 & 17.28 \pm 11.78 & 7.08 \pm 4.79 & 3.01 \pm 2.02 & -42.3 \pm 26.94 & 11.48 \pm 6.83 \\ 2.43 \pm 0.89 & 7.14 \pm 3.09 & 6.11 \pm 2.37 & 6.62 \pm 2.24 & 16.39 \pm 7.84 & -38.69 \pm 5.39 \end{bmatrix}$ | | | | | |

Table 10: Comparison of intensity matrices for the ADP dataset. The time scales are in nanoseconds.

|  | PROBABILITY PER STATE | | | | | |
|---|---|---|---|---|---|---|
|  | I | II | III | IV | V | VI |
| VAMPNETS | 0.30 | 0.24 | 0.20 | 0.15 | 0.11 | 0.01 |
| NEURALMJP | 0.30 | 0.31 | 0.23 | 0.10 | 0.05 | 0.01 |
| FIM-MJP 1% NOISE | 0.28 | 0.28 | 0.24 | 0.07 | 0.10 | 0.03 |
| FIM-MJP 10% NOISE | 0.30 | 0.30 | 0.31 | 0.06 | 0.01 | 0.02 |

Table 11: Comparison of the stationary distribution on the ADP dataset of FIM-MJP, VAMPnets Mardt et al. (2017) and NeuralMJP (Seifner and Sánchez, 2023). The states are ordered such that the protein conformations associated to a given state are comparable in both models. We use the labels of NeuralMJP to evaluate FIM-MJP.

|  | RELAXATION TIME SCALES (IN $ns$) | | | | |
|---|---|---|---|---|---|
| VAMPNETS | 0.008 | 0.009 | 0.055 | 0.065 | 1.920 |
| GMVAE | 0.003 | 0.003 | 0.033 | 0.065 | 1.430 |
| MSM | - | - | - | - | 1.490 |
| NEURALMJP | 0.009 | 0.009 | 0.043 | 0.069 | 0.774 |
| FIM-MJP 1% NOISE (NEURALMJP LABELS) | 0.008 | 0.009 | 0.079 | 0.118 | 0.611 |
| FIM-MJP 10% NOISE (NEURALMJP LABELS) | 0.007 | 0.011 | 0.019 | 0.038 | 0.091 |
| FIM-MJP 1% NOISE (KMEANS LABELS) | 0.001 | 0.003 | 0.046 | 0.142 | 0.455 |
| FIM-MJP 10% NOISE (KMEANS LABELS) | 0.002 | 0.004 | 0.018 | 0.034 | 0.070 |

Table 12: Relaxation time scales for six-state Markov models of ADP. The time scales are ordered by size and reported in nanoseconds. VAMPnet results are taken from Mardt et al. (2017), GMVAE from Varolgünes et al. (2019), MSM from Trendelkamp-Schroer and Noé (2014) and NeuralMJP from (Seifner and Sánchez, 2023).

### F.3 Ion Channel

We consider the 1s observation window that has been used in (Köhs et al., 2021) and (Seifner and Sánchez, 2023) and split it into 50 paths of 100 points. This dataset was provided to us via private communication. We then apply a Gaussian Mixture Model (GMM) to classify the experimental data into discrete states as shown in figure 10.

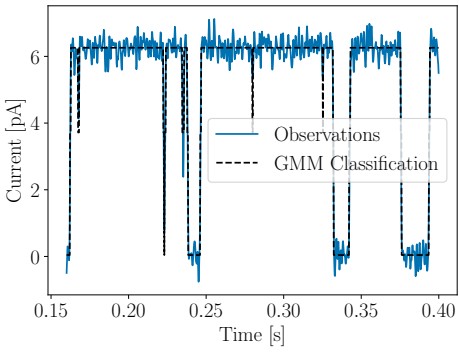

Figure 10: Classification of the ion channel dataset into states.

The predictions of our models and NeuralMJP are shown in table 13. Table 14 reports the stationary distributions and Table 15 reports the mean first-passage times.

| Model | Intensity Matrix |
|---|---|
| NeuralMJP | $\begin{bmatrix} -57.73 & 55.81 & 1.93 \\ 102.13 & -306.93 & 204.81 \\ 0.70 & 26.05 & -26.75 \end{bmatrix}$ |
| FIM-MJP 1% Noise (NeuralMJP Labels) | $\begin{bmatrix} -64.65 & 62.25 & 2.40 \\ 110.55 & -334.05 & 223.50 \\ 0.78 & 31.53 & -32.30 \end{bmatrix}$ |
| FIM-MJP 10% Noise (NeuralMJP Labels) | $\begin{bmatrix} -92.63 & 85.83 & 6.79 \\ 49.31 & -141.72 & 92.40 \\ 2.86 & 32.72 & -35.58 \end{bmatrix}$ |
| FIM-MJP 1% Noise (GMM Labels) | $\begin{bmatrix} -116.37 & 114.65 & 1.73 \\ 271.88 & -716.52 & 444.64 \\ 0.56 & 49.69 & -50.25 \end{bmatrix}$ |
| FIM-MJP 10% Noise (GMM Labels) | $\begin{bmatrix} -104.01 & 97.30 & 6.71 \\ 82.72 & -215.58 & 132.86 \\ 2.89 & 40.29 & -43.18 \end{bmatrix}$ |

Table 13: Comparison of intensity matrices for the ion channel dataset. We cannot report error bars here because the dataset is so small that it gets processed in a single batch.

| | BOTTOM | MIDDLE | TOP |
|---|---|---|---|
| KÖHS ET AL. (2021) | 0.17961 | 0.14987 | 0.67052 |
| NEURALMJP (1 SEC) | 0.17672 | 0.09472 | 0.72856 |
| FIM-MJP 1% NOISE (NEURALMJP LABELS) | 0.18224 | 0.10156 | 0.71621 |
| FIM-MJP 10% NOISE (NEURALMJP LABELS) | 0.14229 | 0.23090 | 0.62682 |
| FIM-MJP 1% NOISE (GMM LABELS) | 0.19330 | 0.08124 | 0.72546 |
| FIM-MJP 10% NOISE (GMM LABELS) | 0.17348 | 0.19610 | 0.63042 |

Table 14: Stationary distribution for the switching ion channel process when trained on the one-second window.

| | KÖHS ET AL. (2021) | | | NEURALMJP | | | FIM-MJP 1% NOISE (NEURALMJP LABELS) | | |
|---|---|---|---|---|---|---|---|---|---|
| $\tau_{ij}/s$ | BOTTOM | MIDDLE | TOP | BOTTOM | MIDDLE | TOP | BOTTOM | MIDDLE | TOP |
| BOTTOM | 0. | 0.068 | 0.054 | 0. | 0.019 | 0.031 | 0 | 0.017 | 0.027 |
| MIDDLE | 0.133 | 0. | 0.033 | 0.083 | 0. | 0.014 | 0.068 | 0 | 0.012 |
| TOP | 0.181 | 0.092 | 0. | 0.119 | 0.038 | 0. | 0.098 | 0.031 | 0 |
| | FIM-MJP 10% NOISE (NEURALMJP LABELS) | | | FIM-MJP 1% NOISE (GMM LABELS) | | | FIM-MJP 10% NOISE (GMM LABELS) | | |
| $\tau_{ij}/s$ | BOTTOM | MIDDLE | TOP | BOTTOM | MIDDLE | TOP | BOTTOM | MIDDLE | TOP |
| BOTTOM | 0 | 0.013 | 0.026 | 0 | 0.009 | 0.016 | 0 | 0.011 | 0.022 |
| MIDDLE | 0.063 | 0 | 0.016 | 0.036 | 0 | 0.007 | 0.045 | 0 | 0.013 |
| TOP | 0.086 | 0.029 | 0 | 0.055 | 0.02 | 0 | 0.065 | 0.024 | 0 |

Table 15: Mean first-passage times of the predictions of various models on the Switching Ion Channel dataset. We compare against (Köhs et al., 2021) and NeuralMJP (Seifner and Sánchez, 2023). Entry $j$ in row $i$ is mean first-passage time of transition $i \rightarrow j$ of the corresponding model.

## F.4 Discrete Flashing Ratchet

We use the same datasets that were used in Seifner and Sánchez (2023), which contains 5000 paths on grids of size 50 that lie between times 0 and 2.5. This dataset was provided to us via private communication. We used 4500 paths to evaluate our model. The predicted intensity matrices for the DFR and the ground truth are shown in table 16.

| Model | Intensity Matrix |
|---|---|
| Ground Truth | $\begin{bmatrix} -1.97 & 0.61 & 0.37 & 1 & 0 & 0 \\ 1.65 & -3.26 & 0.61 & 0 & 1 & 0 \\ 2.72 & 1.65 & -5.37 & 0 & 0 & 1 \\ 1 & 0 & 0 & -3 & 1 & 1 \\ 0 & 1 & 0 & 1 & -3 & 1 \\ 0 & 0 & 1 & 1 & 1 & -3 \end{bmatrix}$ |
| FIM-MJP 1% Noise | $\begin{bmatrix} -1.88 \pm 0.09 & 0.52 \pm 0.06 & 0.31 \pm 0.05 & 0.99 \pm 0.09 & 0.03 \pm 0.00 & 0.03 \pm 0.01 \\ 1.62 \pm 0.12 & -3.34 \pm 0.13 & 0.57 \pm 0.10 & 0.06 \pm 0.01 & 1.04 \pm 0.14 & 0.05 \pm 0.01 \\ 2.73 \pm 0.31 & 1.66 \pm 0.19 & -5.60 \pm 0.55 & 0.12 \pm 0.03 & 0.10 \pm 0.01 & 1.00 \pm 0.27 \\ 0.97 \pm 0.10 & 0.05 \pm 0.01 & 0.04 \pm 0.01 & -3.02 \pm 0.17 & 0.99 \pm 0.10 & 0.97 \pm 0.09 \\ 0.05 \pm 0.01 & 0.98 \pm 0.12 & 0.05 \pm 0.01 & 0.95 \pm 0.15 & -3.05 \pm 0.18 & 1.01 \pm 0.11 \\ 0.07 \pm 0.02 & 0.05 \pm 0.01 & 0.96 \pm 0.11 & 0.94 \pm 0.10 & 1.03 \pm 0.11 & -3.05 \pm 0.19 \end{bmatrix}$ |
| FIM-MJP 10% Noise | $\begin{bmatrix} -1.61 \pm 0.10 & 0.46 \pm 0.07 & 0.23 \pm 0.05 & 0.88 \pm 0.10 & 0.02 \pm 0.00 & 0.02 \pm 0.00 \\ 1.42 \pm 0.11 & -2.78 \pm 0.13 & 0.48 \pm 0.09 & 0.04 \pm 0.01 & 0.81 \pm 0.12 & 0.04 \pm 0.01 \\ 2.68 \pm 0.34 & 1.47 \pm 0.17 & -4.93 \pm 0.49 & 0.06 \pm 0.01 & 0.06 \pm 0.01 & 0.65 \pm 0.25 \\ 0.87 \pm 0.12 & 0.03 \pm 0.01 & 0.03 \pm 0.00 & -2.53 \pm 0.20 & 0.80 \pm 0.09 & 0.80 \pm 0.10 \\ 0.04 \pm 0.01 & 0.84 \pm 0.12 & 0.03 \pm 0.00 & 0.84 \pm 0.17 & -2.61 \pm 0.19 & 0.87 \pm 0.10 \\ 0.05 \pm 0.01 & 0.03 \pm 0.01 & 0.78 \pm 0.09 & 0.86 \pm 0.09 & 0.93 \pm 0.12 & -2.65 \pm 0.15 \end{bmatrix}$ |

Table 16: Comparison of intensity matrices for the DFR dataset on the irregular grid.

### F.5 Modeling Protein Folding through Bistable Dynamics

The work of Mardt et al. (2017) introduces a simple protein folding model via a $10^5$ step trajectory simulation in a 5-dimensional Brownian dynamics framework, governed by:

$$dx(t) = -\nabla U(x(t)) + \sqrt{2}dW(t) \quad,$$

with the potential $U(x)$ being dependent solely on the norm $r(x) = |x|$ as follows:

$$U(x) = \begin{cases} -2.5[r(x) - 3]^2 & \text{, if } r(x) < 3 \\ 0.5[r(x) - 3]^3 - [r(x) - 3]^2 & \text{, if } r(x) \geq 3 \end{cases}$$

This model exhibits bistability in the norm $r(x)$, encapsulating two states akin to the folded and unfolded conformations of a protein.

We use the dataset of Seifner and Sánchez (2023) and apply a Gaussian-Mixture-Model to classify the dataset into two states. The decision boundary of the classifier seems to be based on the absolute absolute value of the radius, namely the classifier seems to classify all states with a radius smaller than approximately 2 into the lower state (see figure 11).

Seifner and Sánchez (2023) generated 1000 trajectories, each with 100 steps after a 1000-step burn-in period. We used 900 paths to evaluate our model. The results are shown in table 17. Table 18 compares the stationary distributions obtained from our models to the ones from Mardt et al. (2017) and Seifner and Sánchez (2023).

|  | LOW STD | HIGH STD |
|---|---|---|
| MARDT ET AL. (2017) | 0.73 | 0.27 |
| NEURALMJP | 0.74 | 0.26 |
| FIM-MJP 1% NOISE (NEURALMJP LABELS) | 0.73 | 0.27 |
| FIM-MJP 10% NOISE (NEURALMJP LABELS) | 0.62 | 0.38 |
| FIM-MJP 1% NOISE (GMM LABELS) | 0.70 | 0.30 |
| FIM-MJP 10% NOISE (GMM LABELS) | 0.65 | 0.35 |

Table 18: Stationary distribution of the model predictions on the protein folding dataset

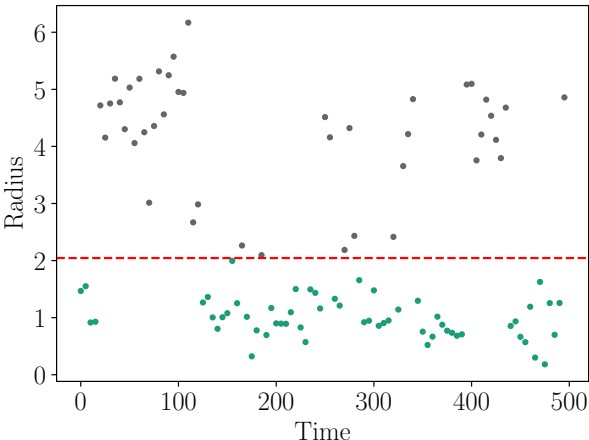

Figure 11: Classification of the protein folding dataset into a Low and a High state. The GMM-Classifier has learned a decision boundary close to the radius 2.

|  | LOW STD → HIGH STD | HIGH STD → LOW STD |
|---|---|---|
| NEURALMJP | 0.028 | 0.085 |
| FIM-MJP 1% NOISE (NEURALMJP LABELS) | $0.019 \pm 0.003$ | $0.054 \pm 0.011$ |
| FIM-MJP 10% NOISE (NEURALMJP LABELS) | $0.034 \pm 0.005$ | $0.055 \pm 0.008$ |
| FIM-MJP 1% NOISE (GMM LABELS) | $0.054 \pm 0.005$ | $0.154 \pm 0.018$ |
| FIM-MJP 10% NOISE (GMM LABELS) | $0.050 \pm 0.006$ | $0.093 \pm 0.011$ |

Table 17: Predicted transition rates on the protein folding dataset

## F.6 A Toy Two-Mode Switching System

In their study, Köhs et al. (2021) produced a time series derived from the trajectory of a switching stochastic differential equation

$$dy(t) = \alpha_{z(t)}(\beta_{z(t)} - y(t)) + 0.5dW(t),$$

with parameters $\alpha_1 = \alpha_2 = 1.5$, $\beta_1 = -1$, and $\beta_2 = 1$. For a concise overview of the generation process, the reader is directed to (Köhs et al., 2021) for comprehensive details. We use the same dataset that was generated in (Seifner and Sánchez, 2023) using the code of (Köhs et al., 2021) which contains 256 paths of length 67 to evaluate our model. Our results are shown in table 20.

|  | BOTTOM → TOP | TOP → BOTTOM |
|---|---|---|
| GROUND TRUTH | 0.2 | 0.2 |
| KÖHS ET AL. (2021) | 0.64 | 0.63 |
| NEURALMJP | 0.19 | 0.36 |
| FIM-MJP 1% NOISE | 0.43 | 0.25 |
| FIM-MJP 10% NOISE | 0.23 | 0.15 |

Table 19: Two-Mode Switching System transition rates. We do not report error bars here because the dataset is so small that it runs in a single batch.

## F.7 Initial Distributions

For completeness, we report in this section the initial distributions predicted by FIM-MJP on various datasets as well as the heuristic initial distribution (which is computed simply by counting the number of state occurrences at the first observation). We observe that FIM-MJP typically captures the initial distribution quite well. An exception is the Two-Mode Switching System for which FIM-MJP falsely predicts a non-zero probability of the first state. This might happen because we did not capture this case in our training distribution which could be an improvement for future work.

| DATASET | PREDICTED $\pi_0$ | HEURISTIC $\pi_0$ |
|---|---|---|
| DFR $V = 1$ | $[0.22, 0.15, 0.11, 0.19, 0.16, 0.17]$ | $[0.3, 0.14, 0.06, 0.2, 0.17, 0.14]$ |
| IONCH | $[0.14, 0.11, 0.75]$ | $[0.24, 0.08, 0.68]$ |
| ADP | $[0.34, 0.3, 0.23, 0.11, 0.02, 0.0]$ | $[0.33, 0.29, 0.25, 0.11, 0.02, 0.0]$ |
| TWO-MODE SYSTEM | $[0.32, 0.68]$ | $[0.0, 1.0]$ |
| PROTEIN FOLDING | $[0.75, 0.25]$ | $[0.73, 0.27]$ |

Table 20: Comparison of the predicted initial distribution of the model versus the heuristic initial distribution of various datasets.

