# OpenReview forum: "Foundation Inference Models for Markov Jump Processes"
_NeurIPS.cc/2024/Conference — NeurIPS 2024 poster_

### Official Review · Reviewer_pfZc · 2024-07-08

**Soundness:** 3
**Presentation:** 3
**Contribution:** 3
**Rating:** 7
**Confidence:** 2

**Summary:**

The paper explores the possibility of using transformers to directly infer the parameters of a Markov jump process (MJPs) from a noisy time data set obtained from a time-dependent process. They first train the machine to correctly predict the model parameters by using synthetic datasets containing data obtained with different realizations of MJPs defined with different parameter sets. Later, they show that the models trained in this way are nevertheless able to derive the parameters of a real simulated or experimental process, even though they were not trained on this data. They also show that one-shot inference is as good as state-of-the-art methods when trained on the target datasets.

**Strengths:**

The authors present a very simple yet powerful and practical idea of using neural networks to extract information from a time-dependent process. In contrast to traditional unsupervised approaches, whose goal is to learn the model parameters to generate samples that are as similar as possible to the original, their goal is a fully supervised one to correctly predict the model parameters in inverse synthetic experiments. They show that the neural network is not only powerful enough to infer all the parameters occurring in the master equation in controlled experiments, but also that they can later successfully apply these trained models to unseen real-world datasets.

**Weaknesses:**

Despite the originality of the work, there are some aspects that remain unclear to me. I find that important details about how to apply the trained models to real data in practice are missing. I also think the limitations are not properly explored.

**Questions:**

Do they make their predictions on the basis of coarse-grained data? If so, how are these representations obtained? If not, how is the data standardized so that they can transfer the models?

I did not understand if the adjacency matrix is learned or not, and I am a bit confused about the information given to the machine in real datasets if not.

I find that the limitations of this approach are not properly explored or discussed in the text. Are the processes used to generate the synthetic datasets realistic in a general setup? Are they able to describe complex processes like the data generated with the trap model? or just Gaussian processes such as a Brownian motion?

**Limitations:**

I discused my doubts about the limitations above.

---

> ### Author Rebuttal · Authors · 2024-08-07
>
> We would like to thank the reviewer for the helpful comments and questions, which we know will improve the presentation of our work. Below we address each of them.
>
> **@W1** (On how to apply the model): We kindly refer the reader to our general response above. We hope it addresses the reviewer’s comment on the lack of details on how to apply FIM to any dataset. It will be included into our manuscript (see summary above).
>
> **@W2** (On limitations): Please see our response to your Q3 below.
>
> **@Q1** (Do they make their predictions on the basis of coarse-grained (CGR) data): Yes, FIM takes coarse-grained data as input.
>
> As we explained in our introduction (see e.g. lines 49-54 and lines 116-117), our goal is to tackle the classical MJP inference problem on *coarse-grained (i.e. discrete) space*.
>
> In practice, one can obtain these CGRs by e.g. leveraging pretrained models specifically tailored to performing CGR inference in specific datasets or, more naively, by simply using clustering algorithms.
>
> In our experiments we used the CGR inferred by NeuralMJP, for the sake of fairness when comparing against it. We also used naive CGRs obtained via Gaussian mixture models (see e.g. Section 4.2, lines 283-287) and the KMeans clustering algorithm (see e.g. Section 4.3, lines 305-307 and Appendix E.2, lines 674-683). Table 3 in the main text and Tables 12, 15, 17 and 18 in the Appendix contain our estimated observables wrt. the different CGRs we employed for each of our datasets.
>
> Let us finish this response with the following remark. The inference of CGR should be understood as a different problem from that of the MJP inference. Accordingly, it is to be considered outside the scope of the present work (see e.g. our discussion in lines 102-117 in the related work section). To include such an inference into the FIM methodology, would amount to including an emission probability model into the generative model (Eq. 2) of our synthetic training distribution. Effectively, this would be equivalent to replacing our noise distributions with more complex models. We do not pursue this avenue, but the present work serves as a basis for such an extension.
>
> **@Q2** (On the adjacency matrix): The adjacency matrix $\mathbf{A}$ only appears in our model as part of our synthetic training distribution, and it simply defines the level of sparsity of the embedded Markov chain of the MJPs within the training dataset. Accordingly, the adjacency matrix $\textbf{A}$ is *not needed* as input to the model. One only requires the time series of the observation times and the corresponding values (states) in CG space (see also the general reply above for details).
>
> Finally, note that since FIM returns the $C \times C$ inferred rate matrix $\mathbf{\hat F}$ (see e.g. Eq. 5), the target adjacency matrix can be understood as being implicitly inferred by FIM (that is, it can be extracted from the inferred $\mathbf{\hat F}$ matrix).
>
> **@Q3**. (On limitations): The question is twofold. Let us start by answering whether our synthetic datasets are realistic enough to be useful in a general setup. If we interpret this question as whether our synthetic distributions are wide enough to cover empirical path distributions of real world problems, our answer is *yes*. Indeed, we leveraged one and the same pretrained FIM to infer hidden MJPs from 5 very different target datasets. In particular, the switching ion dataset is a real-world, experimental dataset recorded from a viral ion channel, and the ADP dataset is an all-atom simulation of the alanine dipeptide molecule. Both these datasets correspond to very complex dynamical systems. Yet FIM, trained on our heuristically constructed synthetic dataset, is able to make the same inference as the baselines, without the need of any fine-tuning.
>
> The second question concerns the applicability of our pretrained FIM to diffusion processes and to trajectories sampled from the trap model. Traditionally, MJPs and diffusion processes are treated (or better, defined) as different types of (continuous-time) stochastic process, as they have different supports. Indeed, MJPs are defined as processes which take values on discrete sets (like e.g. the integers), whereas diffusion processes are defined as processes which take values in continuous sets (like e.g. the real numbers). Our methodology is designed to tackle the inference of MJP only, and hence cannot be used to infer diffusion processes.
>
> Nevertheless, let us remark, for the sake of completeness, that there are certain limits in which diffusion process can be asymptotically described in terms of MJPs, as first studied by van Kampen and Kubo (see e.g. Gardiner, 2009, chapter 11, or the very recent work of Winkler et al. 2024). These limits involve state spaces that are typically unbounded, and hence are out of the scope of the present work, which focuses, as we stressed in our abstract, on the inference of MJP on bounded state spaces.
>
> Similarly, trap-like models, understood as phenomenological models for relaxation times (as in the mean-field ferromagnetic model of Griffiths et al. 1966, or the model for aging in disordered systems of Bouchaud 1992), can be understood as MJPs on unbounded state spaces (i.e. space of infinitely many metastable states). Our methodology targets the inference of MJP on bounded state spaces and hence cannot be used to model trajectories from trap-like models.
>
> Let us finish this section by kindly referring the reviewer to our response to weakness W2b of Reviewer sVKK, where we discuss the limitations of our methodology, especially with respect to the synthetic training distribution.
>
> *Links to references*:
> - Gardiner (2009): https://link.springer.com/book/9783540707127
> - Winkler et al. (2024): https://arxiv.org/pdf/2405.03549
> - Griffiths et al. (1966): https://journals.aps.org/pr/abstract/10.1103/PhysRev.149.301
> - Bouchaud (1992): https://jp1.journaldephysique.org/articles/jp1/abs/1992/09/jp1v2p1705/jp1v2p1705.html

---

> > ### Comment · Reviewer_pfZc · 2024-08-09
> >
> > I would like to thank the authors for their detailed explanations. My question Q3 was more about the complexity of the free energy landscapes of inference problems that can be addressed. I apologise for my abuse of language. Is the method still reliable when it comes to glassy dynamics or rugged landscapes?

---

> > > ### Author Response · Authors · 2024-08-12
> > >
> > > Let's now finish with some remarks.
> > >
> > > - Let us note that our methodology is suitable for any choice of the coarse graining procedure. The latter is left to the practitioner, for different regions of the energy landscape can be better suited for the state definition, depending on the application at hand.
> > > - The general applicability of our pretrained FIM was demonstrated by inferring MJPs from synthetic, simulated and experimental datasets of very different nature. However, albeit general enough to cover these cases, our **proposed training distribution should be understood as a first example**.  Our main goal was to demonstrate that designing broad synthetic training distributions can be used to train neural network models to perform (amortized) zero-shot inference of MJP in different situations.
> > > - Future research can involve, for example, extending our dataset to include power-law distributed rates into our datasets, and hence extend the applicability of the newly trained FIM to the case discussed above. One can also extend the pretraining dataset to include MJP of larger space state sizes, featuring connecting networks with different distributions.

---

> ### Author Response · Authors · 2024-08-12
>
> As we understand it, an MJP description of glassy dynamics, as done by e.g. Bouchaud (1992), identifies the metastable states of the disordered system in question (i.e. the set of energy minima characterizing the energy landscape) with the states of the MJP. Within this description, the MJP dynamics depends on the shape of the energy landscape, inasmuch as the transition rates between states $i$ and $j$ ($f_{ij}$ in our notation) are characterized by the depth of the *trap* (that is, the height of the barrier between them).
>
> In equations, practitioners typically write  $f_{ij} = \exp(-E_j/T)$, where $E_j$ is the trap depth and $T$ is the temperature.
>
> What matters when applying FIM to such systems is that
> 1. the number of metastable states in the system **is smaller or equal than** $C$ (the size of the largest state space in our dataset, which we set to 6 in our experiments); and that
> 2. the distribution of energy barriers in the system is such that the corresponding transition rate distribution **is within our training dataset**.
>
> Just to give an example, we can assume an exponential energy barrier distribution and find, by using the expression for $f_{ij}$ above, a rate distribution of the form
>
> $p(f) = A  T  f^{T-1}$, for some constant $A$,
>
> which is a power law. However, our prior rate distribution $p_{FIM}(f)$ is a (mixture of) Beta distribution(s). See e.g. line 150 or lines 473-475. Therefore, we expect FIM *trained on our current dataset*, with Beta priors, to perform poorly for glassy systems whose energy landscapes feature exponentially distributed trap depths.
>
> We truly hope that the above answers the reviewer's question.

---

> > ### Comment · Reviewer_pfZc · 2024-08-13
> >
> > I want to thank again the authors for the detailed explanation. That's what I expected. I think the issue should be discussed in the limitations section. Trap like dynamics are relevant in the context of disordered proteins, as the random energy model presents trap dynamics.
> >
> > I will update my score to accept.

---

> > > ### Author Response · Authors · 2024-08-13
> > >
> > > We would like to thank the reviewer again for their comment and respond that we will include the discussion above into the Limitations section, as suggested.

---

### Official Review · Reviewer_5km2 · 2024-07-13

**Soundness:** 4
**Presentation:** 4
**Contribution:** 4
**Rating:** 8
**Confidence:** 4

**Summary:**

The authors propose a foundation model for Markov jump processes (MJPs) that is trained on sequences drawn from synthetic MJPs to predict the corresponding rate matrix and initial state distribution. High-level arguments are presented for the justification of model trained in such a way to be able to generalize to unseen sequences drawn from different processes. Extensive experimental results are shown demonstrating the model's performance after initial training on synthetic data and without any further fine-tuning.

**Strengths:**

The paper is very clearly written and the approach is motivated well. The lack of finetuning in the experiments showcase well the generalizability of the proposed foundation model. Many experiments are showcased that do a good job of demonstrating the intended strengths of the model.

**Weaknesses:**

As pointed out in the paper, the model naturally has weaknesses generalizing to data that lies outside of the initially exposed distribution that it was exposed to in training, such as on sequences with very high rates or with states that lie outside the initial support.

Outside of this, I could not find any other weaknesses of the paper; I found it to be very good in general.

**Questions:**

I personally believe the success demonstrated by the foundation model with how well it generalizes being due to the simplicity (not a bad thing!) of the data, i.e., single dimensional across time and is Markov. Would you agree with this or am I missing something?

Additionally, I was wondering if "zero-shot" is really appropriate to use here. Typically, zero-shot is used when trying to produce predictions when there is no labeled data to learn from, i.e., predicting "dogs" vs. "cats" with an image model that has seen neither during training. This is commonly achieved by providing some side-information, such as a description of a class. Here the model does indeed not see any "labeled" rate matrices; however, it does see many different sequences realized from this matrix. I see it as less zero-shot and moreso that the inference problem has been amortized. By this, I mean that other methods directly learn a MJP whereas this predicts the parameters to one given a batch of sequences. I am not saying any of this to detract from the paper, but rather just to dial down how to actually think about what it is doing and the language used to describe it. Any thoughts on this would be great to hear.

**Limitations:**

The authors adequately describe the limitations.

---

> ### Author Rebuttal · Authors · 2024-08-07
>
> We thank the reviewer  for both the detailed review and the kind words about our work. Below we address each of the comments and suggestions.
>
> **@Q1** (FIM success due to simplicity of the data): We completely agree with the reviewer. Indeed, we argued in Section 3, lines 119-127 of our paper, that the main assumption underlying our FIM methodology is that *the space of realizable MJPs, which take values on bounded state spaces that are not too large, is simple enough to be covered by a heuristically constructed synthetic distribution over noisy and discretely observed MJPs*.
>
> That being said, we are currently working on extending our methodology to other families of Markov processes,  like e.g. single MJPs on unbounded state spaces, or multidimensional (i.e. coupled) MJPs, for which, despite their significantly larger state space, we think we can define, via information theoretic arguments, synthetic distributions that cover well their realizable trajectories.
>
> **@Q2** (On the zero-shot terminology): We understand the point raised by the reviewer. We use the zero-shot terminology in the spirit of the work in Larochelle et al. (2008). That is, for us, zero-shot learning aims to recognize objects (e.g. stochastic processes) whose instances (e.g. samples or paths) may not have been seen during training.
>
> As the reviewer writes, FIM does indeed not see any labels (i.e. rate matrices) during inference. That is one of our first motivations to use the zero-shot terminology. We understand that what we propose is, perhaps, an extension of this terminology to the case of inference of stochastic processes, which we intended to go hand in hand with our foundation (inference) model terminology (which is also an extension of the nlp terminology).
>
> Nevertheless, we do not want to create any confusion as to how FIM works. We completely agree with the reviewer in that our methodology can be fully understood as an amortized inference procedure. In fact, reviewer EGs8 also drew our attention to the work of Paige&Wood (2018), who described a procedure to train a recognition model offline via amortized inference.
>
> What we therefore propose is to
> - first, add to the related work section a paragraph on amortized inference and offline learning, where we reference Paige&Wood (2018), together with other representative works in this direction, and connect them with our approach; and
> - second, explain in Section 3 what our motivation for the zero-shot terminology is, and how we understand it as an amortized inference procedure.
>
> Does the reviewer think that these modifications will help avoid misunderstandings with our terminology?
>
> *Links to references:*
>
> - Larochelle et al. (2008): https://cdn.aaai.org/AAAI/2008/AAAI08-103.pdf
> - Paige&Wood (2018): https://arxiv.org/pdf/1602.06701

---

> > ### Comment · Reviewer_5km2 · 2024-08-10
> >
> > Thank you for the in depth responses to my questions!  I do in fact believe this extended discussion over the zero-shot terminology would go a long ways towards better communicating this concept to the reader.
> >
> > I maintain my original score.

---

### Official Review · Reviewer_sVKK · 2024-07-15

**Soundness:** 3
**Presentation:** 3
**Contribution:** 3
**Rating:** 6
**Confidence:** 4

**Summary:**

This study describes a foundation model for a specific stochastic process, called Markov Jump Process (MJP). The foundation model, called FIM, is trained with a large number of synthetically generated MJP data set. It is shown that the pre-trained FIM can make a zero-shot inference. The capability of FIM is demonstrated by using a few data set.

**Strengths:**

The paper is well written. It clearly formulates the problem setup and the proposed approach, which makes it read well. While the proposed method is exploratory, it demonstrates some interesting capability. The numerical experiments are limited, but well documented.

**Weaknesses:**

The numerical experiments are not strong enough. Numerical experiments should achieve either 1) comparing the performance of the proposed model against a cohort of the state-of-the-art methods, or 2) to systematically demonstrate the capability and limitations of the proposed model and investigate the behavior of the model under a range of important conditions. However, the manuscript does not achieve both. Based on the Appendix, I believe that the authors would have performed an extensive number of studies. A more detailed analysis of the model behaviors with respect to the range of parameters will be appreciated.

**Questions:**

1. It is not clearly explained how the data are scaled. Since the training data is generated from equations, it should be fine. But for an inference the input data should be correctly scaled. How is the inference data scaled? Are there any effects of the scaling on the inference result?

2. Similar to question 1, how the time scale is determined. The model predicts "rate", which depends on the time scale of the data. What method do you use for the inference time? Is the time scaling method general enough to be applied for any real-life MJP?

3. A more detailed explanation about the number of states is required. The model is trained with the maximum number of states of six. What happens when the number of states is less than 6? What do $A$ and $F$ look like? Do the redundant elements of the matrices become exactly zero? If not, how do you normalize the transition probability?

4. Similar to 3, it is interesting to see how the model behaves if the number of states is larger than the maximum number of states in the training, as in real-life problems it is difficult to know or restrict the number of states.

5. During the training, when the number of states is less than six, how do you compute $Var\hat{f}_{ij}$? Note that $Var\hat{f}_{ij}$ is used in the denominator and in a log.

6. How accurate is the prediction of the initial state, $\pi_0$?

7. The prediction requires a prior distribution for $A$. What are the effects of $A$ on the prediction? What happens if the data is generated from a MJP with a different distribution of $A$?

8. The inference of FIM requires a fairly large amount of input data. A conventional approach is to fit a state-space model with the same amount of data to learn the MJP parameters and make an inference. How does FIM compare against the standard ML approach?

---

> ### Author Rebuttal · Authors · 2024-08-07
>
> Let us thank the reviewer for the detailed review, as well as for the proposed questions and weaknesses. However, let us remark here that some of the proposed weaknesses are somewhat vague. Accordingly, below we ask the reviewer to be more specific in (some of) their remarks.
>
> **@W1** (On comparisons against state-of-the-art): We tackle the problem of MJP inference from noisy data for processes defined on bounded state spaces. A prominent application of these processes is the modeling of conformational dynamics in proteins and molecules, and in our paper we considered two such examples: the conformational switching of ion channels and the molecular dynamics of a 22-atom molecule. We compared FIM against
>
> - The VampNets model of Mardt et al. (2017): the go-to, state-of-the-art deep-learning approach for modeling molecular dynamics via discrete-time Markov chains;
> - The SDiff model of Koehns et al. (2021): a recent variational model that has been applied to the modeling of switching ion channels; and
> - The NeuralMJP model of Seifner & Sanchez (2023): to the best of our knowledge, the most recent, state-of-the-art neural network model available online, which leverages neural variational inference and neural ODEs to infer hidden MJPs.
>
> We therefore believe that we have compared our methodology against a set of state-of-the-art models. Nevertheless, we would be very happy if the reviewer could point out any state-of-the-art model we did not include into our related work.
>
> **@W2a** (On the systematic demonstration of the capabilities of FIM): We have empirically demonstrated that one and the same pretrained FIM can be used to estimate stationary distributions, relaxation times, mean first-passage times, time-dependent moments and thermodynamic quantities (i.e. the entropy production) from five very different, noisy and discretely observed MJPs, taking values in state spaces of different dimensionalities, without the need of any fine-tuning. What is more, FIM was shown to perform on par with all the baselines.
>
> We believe that these tests cover well the capabilities of FIM. We therefore would very much appreciate if the reviewer could be more specific as to what other demonstrations they are referring to.
>
> **@W2b** (On the systematic demonstration of the limitations of FIM): As we explained in the limitations section of our paper, the main limitations of FIM are related to the synthetic training distribution.  We expect that if one leverages FIM to infer the rate matrix from an empirical process whose path distribution lies well outside our synthetic training distribution, one will obtain poor rate estimates. Figure 4 represents such an example.
>
> Beyond this, we have also explored how other features of our synthetic training distribution affect FIM. Indeed, as we explained in lines 199-207 and lines 212-217 of the main text, FIM is expected to perform best for the context number $c(300, 100)$, a number which is specified by the training distribution. We studied how FIM handles context numbers that are different from the optimum in both Appendix D.1 and D.4, and we refer the reviewer to them for details.
>
> Similarly, we have also studied the effect of training FIM on a synthetic dataset which contains only six-state MJPs in Appendix D.3.
>
> Again, we believe all these experiments cover well the limitations and features of FIM, especially as regards the synthetic training distribution, and we would be glad if the reviewer could be more specific as to which type of experiments they refer to.
>
> **@Q1, Q2, Q3**: We kindly refer the reviewer to our general comment above.
>
> **@Q4**: As explained in the general comment above, FIM always returns a $C\times C$ matrix. It is therefore not possible for it to predict intensity matrices of dimension higher than C.
>
> **@Q5**: As can be read from equation 6, during training we ask FIM to predict zeros for those redundant matrix elements. Indeed, the first two terms in equation 6 are both multiplied by $a_{ij}$, which is zero when the target rate is zero. Similarly, the second two terms are multiplied by $(1-a_{ij})$, so that only these terms are active when the target rate is zero.
>
> **@Q6**: Let us answer this question by reporting our predictions for the DFR dataset with $V=1$. The ground-truth initial condition is given by the distribution
>
> $\pi_0 = [0.301, 0.137, 0.062, 0.200, 0.159, 0.141]$.
>
> FIM predicts
>
> $\hat \pi_0 = [0.224, 0.147, 0.112, 0.189, 0.155, 0.173]$.
>
> Our estimates are therefore reasonably accurate. For completeness we will report both our estimated $\hat \pi_0$ and their empirical counterpart for all target datasets in the Appendix.
>
> **@Q7**: The adjacency matrix $\textbf{A}$ only appears as part of our training distribution, and it simply defines the level of sparsity of the embedded Markov chain of the MJPs within the training dataset. Thus, no information about the adjacency matrix is needed during inference.
>
> As explained in Appendix B.1, lines 479-484, the adjacency matrix is sampled from an Erdős–Rényi model, with edge probability 0.5, and is rejected if the corresponding graph is not connected. This prior on $A$ is, just as all other priors in Eq. 2, part of our definition of the FIM synthetic training distribution. Our experiments corroborate that these heuristic priors, which we define in Appendix B.1, are expressive enough for our trained FIM model to perform well in our 5 target datasets.
>
> **@Q8**: In our experiments we make use of as many data points as the baselines in all experiments, for fairness in comparison. We give the sizes of each of these datasets in each subsection of section 4 in the paper.
>
> We however do not fully understand what the reviewer means by the last question: *How does FIM compare against the standard ML approach?* All our baselines are different instances of the “standard ML approach”. Indeed, the goal of our paper is to show that FIM provides a good alternative to the standard paradigm.

---

> > ### Comment · Reviewer_sVKK · 2024-08-12
> >
> > I would like to thank the author for the clarifications. My last comment is a question regarding a model trained for the target distribution versus a foundation model trained with a large number of data. In the conventional way, you train a MJP  model for a particular data set and make an inference for the data set with the same underlying distribution. In FIM, you train FIM using a large collection of data set and make an inference on a new data set. It would be interesting to compare the accuracy of these two different approaches. It will be surprising if FIM outperforms a MFP model tailored for the particular data generating distribution. But it will be interesting if the FIM can show a comparable accuracy.
> >
> > Re-iterating my first review, the study is not perfect and still exploratory. But it is interesting and certainly has a merit. I will update my score from 5 to 6.

---

> ### Author Response · Authors · 2024-08-07
>
> *Links to references*:
> - Mardt et al.(2017): https://www.nature.com/articles/s41467-017-02388-1
> - Koehns et al. (2021): https://proceedings.neurips.cc/paper/2021/hash/abec16f483abb4f1810ca029aadf8446-Abstract.html
> - Seifner & Sanchez (2023): https://proceedings.mlr.press/v202/seifner23a.html
> - Gazzarrini et al. (2006): https://www.pnas.org/doi/full/10.1073/pnas.0600848103

---

> ### Author Response · Authors · 2024-08-13
>
> We would like to thank the reviewer for their comment and respond that what the reviewer asks is precisely what we did.
>
> Our three baselines (NeuralMJP, SDiff, and VampNets) were trained in the conventional way. That is, our three baseline models were trained on the target datasets. We mention this in our abstract and again on line 82. See also the discussion on lines 234, 237, and 238.
>
> We realize, however, that we may not have been that clear about this. Therefore, to make this point more explicit, we will add a line in the Experiments section to clarify that all baselines are trained on the target datasets.

---

### Official Review · Reviewer_EGs8 · 2024-07-16

**Soundness:** 3
**Presentation:** 3
**Contribution:** 3
**Rating:** 7
**Confidence:** 3

**Summary:**

This work presents a framework for amortizing inference on Markov Jump Processes by learning a foundational model in a supervised fashion from synthetic data. Once learned, the "foundation model" is shown to be successful at zero-shot inference in MJP across a range of domains, out-performing SOTA models that are fine-tuned to the target datasets.

**Strengths:**

Clear exposition of why problem is important and the motivation behind the approach. Clear exposition of the method and results.

**Weaknesses:**

Discussion of prior literature could be improved. Some relevant work is on amortized inference in other types of Bayesian models. E.g. "Inference Networks for Sequential Monte Carlo in Graphical Models" learns a supervised model on synthetic data for zero shot inference on Bayesian regression coefficients.

**Questions:**

You mention a main limitation is that generalization is poor outside of the family of models assumed in the synthetic training data. What happens to generalization as you scale up the NN model and widen the family of MJPs in the training data (like for LLMs)?

**Limitations:**

Limitations are adequately discussed.

---

> ### Author Rebuttal · Authors · 2024-08-07
>
> First of all, we would like to thank the reviewer for taking the time to review our work and for the helpful comments and questions.
>
> **@W1**: We thank the reviewer for pointing this reference out. We will include it into our related work section, together with other representative references on amortized inference and offline learning.
>
> **@Q1**: Indeed, we expect that if one leverages FIM to infer the rate matrix from an empirical process whose path distribution lies well outside our synthetic training distribution, one will obtain poor rate estimates.  As we commented in the limitation section, Figure 4 represents such an example.
>
> As implied by the reviewer, there are two paths one can follow to improve the performance of FIM in such cases:
>
> (i) scaling the parameter count of FIM, and
>
> (ii) widening the MJP training distribution.
>
> *Regarding parameter scalings*, we have empirically observed that, for our current 45K MJP synthetic dataset, increasing the parameter count does not necessarily improve the performance of FIM. This can be seen, for example, in Figure 7 of Appendix D.2, in which, broadly speaking, the performance of different FIM architectures, with respect to our synthetic distribution, is statistically similar. We invite the reviewer to read our discussion in lines 592-606 for details.
>
> *Regarding the MJP training distribution*, we do expect that widening our synthetic training distribution will improve the performance of FIM in e.g. the cases reported in Fig. 4. Future work will explore how to define wider synthetic distributions that contain rare or exceptional processes, and how scaling of the FIM parameter count in such cases affects their encoding.

---

> > ### Comment · Reviewer_EGs8 · 2024-08-13
> > **Response to Rebuttal**
> >
> > I thank the authors for their thoughtful response to my review and am glad to keep my score as accept. I look forward to future work in this area building upon this paper.

---

### Author Rebuttal · Authors · 2024-08-07

We would like to thank all reviewers for their valuable comments and questions. We have carefully read and addressed all of them, for each reviewer separately, in the discussions below. We have however noticed a couple of common questions among (some of) the reviewers, namely

1. How is the input data of FIM scaled, so that FIM can be applied to real-world data?
2. How does FIM handle MJPs whose state space size is smaller than the maximum size $C$ in the training ensemble?

Let us address each of these, and summarise the main updates we will do to the manuscript, which we believe answer all questions/concerns of the reviewers, here.

**1. On the scaling of the input data**: As we explained in Appendix C.2, lines 549-556, FIM takes as input:

- the normalized sequence of observation times, which lie on the interval $[0, 1]$;
- the values of the process at those observation times, which are sequences of integers taken from the set $\{0, 1, …, C-1\}$, where $C$ is the size of the largest state space in our training dataset;
- the number of observations per simulation; and
- the time scale $T^*$, which we define as the observation time of the last event in the (unnormalized) input time series.

(*On the scaling of the observation values*).  The observation values do not need to be scaled since they are nothing but sequences of integers from the set $\{0, 1, 2, … C-1 \}$. Indeed, as we explained in our introduction (see e.g. lines 49-54 and lines 116-117), our aim is to tackle the classical MJP inference problem on coarse-grained (i.e. discrete) space.

(*On the scaling of the observation times*). In order to be able to process arbitrary input data, FIM requires the observations times to lie on the interval $[0, 1]$. Indeed, as we explained in Appendix B.1, lines 485-490 and lines 505-507, our target rate matrices take, by construction, values between 0 and 1 only. We use these matrices to simulate MJPs on the time interval [0, 10], and then map our random observation times to the interval [0, 1]. We do this mapping by normalizing all observation times wrt. the maximum observation time in the set of paths we simulate per MJP. That is, we define the *time scale* of the simulated paths to be the time $T^*< 10$ of the last event in the path set of a given MJP. Naturally, this normalization affects the target rates (which have units of one over time) only by the factor $T^*$. *FIM is trained to predict the normalized rates*.

The true (unnormalized) rates can then simply be computed by multiplying the matrix $\mathbf{\hat F}$ returned by FIM with $1/T^*$. In practice, this normalization happens inside FIM, which is why the model requires $T^*$ as input.

Note that, as we empirically demonstrated in the paper, this rescaling procedure allows us to work with real-world MJPs of arbitrary time scales. For example, the time scales for the switching ion channel dataset were more than 500 times smaller than the time scales in our training dataset.

**2. FIM and MJP with different state space sizes**. As we mentioned above, FIM takes as input integer sequences (i.e. the observations) on the discrete set $\{0, 1, 2, … C-1 \}$. FIM always returns the $C(C-1)$ off-diagonal elements of the $C \times C$ inferred rate matrix $\mathbf{\hat F}$, whose diagonal elements are then computed as $\hat F_{ii} = \sum_{j \neq i} \hat F_{ij}$.

In our experiments we trained FIM on a synthetic dataset with $C=6$.

Now, we arrange all the target rate matrices $\mathbf{F}$ within our training dataset for MJPs with state spaces of size $c < C=6$, to be the leftmost block diagonal $c \times c$ matrix within a $6\times 6$ matrix of zeros, so that the redundant matrix elements are always zero. As can be read from equation 6, and as we explained in lines 196-198, we train FIM to predict zeros for those redundant matrix elements.

In practice, however, our trained FIM does not exactly predict zeros for those redundant matrix elements. In our experiments the user knows a priori the number of states of the hidden process, so we explicitly set the redundant matrix elements to zero, and only then compute the corrected diagonal (i.e. the normalization) of the output rate matrix. Note that this assumption is typically made by the baselines (see e.g. Seifner & Sanchez, 2023).

To illustrate that FIM nevertheless predicts very small values for the entries of the rate matrix which are supposed to be zero, we considered again the 3-state switching ion dataset, and computed again the stationary distribution from the complete $6\times 6$ output matrix, without zeroing out the elements outside the $3\times 3$ leftmost block diagonal matrix. We obtained the following:

$
\hat p^* = (0.154, 0.226, 0.590, 0.010, 0.008, 0.011).
$

Note that, as expected, the last three entries have very small probabilities. Also note that the first three entries agree well with the results reported in Table 3.

**SUMMARY OF MAIN MODIFICATIONS**:

- We will add to the related work section a paragraph on amortized inference and offline learning, where we reference the work of Paige&Wood (2018), together with other representative works in this direction (Weakness 1, Reviewer EGs8);
- We will explain in Section 3 what our motivation for the zero-shot terminology is, and how we understand it as an amortized inference procedure (Question 2, Reviewer 5km2);
- We will move the subsection “How to use the model” of Appendix C.2 into a new Appendix section. This new Appendix will gather and extend what we wrote above **on the scaling of the input data** (Question 1 and 2, Reviewer sVKK and Weakness 1, Reviewer pfZc);
- We will include a new Appendix which will contain our notes **on FIM and MJP with different state space sizes** which we wrote above (Question 3, Reviewer sVKK).
- We will report both our estimated initial conditions ($\hat \pi_0$) and their empirical counterpart for all datasets in the Appendix (Question 6, Reviewer sVKK).

---

### Decision · Program_Chairs · 2024-09-25

**Decision:**

Accept (poster)

**Comment:**

The reviewers agree that this is an interesting paper that deserves to be published. A limitation is that this work uses empirical rather than theoretical investigations to demonstrate the usefulness of the proposed idea, and the paper would be strengthened if this experiments more clearly demonstrated the limitations of this methodology. The authors might want to think about this (as well as other reviewer comments) when they prepare the final version of this paper.